# A physics-based Antarctic melt detection technique: Combining AMSR-2, radiative transfer modeling, and firn modeling

Marissa E. Dattler[1,2,3], Brooke Medley[3], C. Max Stevens[2,3]

[1]Department of Atmospheric and Oceanic Sciences, University of Maryland, College Park, MD 20740, USA
[2]Earth System Science Interdisciplinary Center, University of Maryland, College Park, MD 20740, USA
[3]NASA Goddard Space Flight Center, Greenbelt, MD 20771, USA

*Correspondence to*: Marissa E. Dattler (marissa.e.dattler@nasa.gov)

**Abstract.** Surface melt on ice shelves has been linked to hydrofracture and subsequent ice shelf breakup. Since the 1990s, scientists have been using microwave radiometers to detect melt on ice shelves and ice sheets by applying various statistical
thresholding techniques to identify significant increases in brightness temperature that are associated with melt. In this study, instead of using a fixed threshold, we force the Snow Radiative Transfer model (SMRT) with outputs from the Community Firn Model (CFM) to create a dynamic, physics-based threshold for melt. In the process, we also combine our method with statistical thresholding techniques and produce microwave grain size information in the process. We run this "Hybrid Method" across the Larsen C ice shelf as well as thirteen sites on the Antarctic Ice Sheet. Melt and non-melt days from the
Hybrid Method and three statistical thresholding techniques match with surface energy balance within 94±1%; the effect of melt on the passive microwave is mostly binary and thus largely detectable by statistical thresholding techniques as well as physics-based techniques. Rather than always replacing statistical thresholding techniques with the Hybrid Method, we recommend using the Hybrid Method in studies where melt volume or grain size is of interest. In this study, we show that the Hybrid Method can be used to (a) model dry snow brightness temperatures of Antarctic snow and (b) derive a measure of
grain size; therefore, it is an important step forwards towards using firn and radiative transfer modeling to quantify melt rather than to simply detect melt days.

## 1 Introduction

Studying Antarctic surface melt is critical for evaluating the relationship between the Antarctic Ice Sheet (AIS) and the
climate system. Climate change has been associated with increased surface melt and pond formation in Antarctica, particularly in the Antarctic Peninsula (Scambos et al., 2000). While only a small percentage of surface meltwater runs off into the ocean, recent studies have shown that meltwater on the surface of ice shelves can lead to hydrofracture and ice shelf collapse (Scambos et al., 2000; Banwell et al., 2013; Kingslake et al., 2017; Bell et al., 2018). Since ice shelves provide a buttressing effect to ice upstream, their collapse can indirectly result in mass loss of the AIS through increased ice discharge
into the ocean contributing to sea level change (Rignot, 2004; Berthier et al., 2012).

To identify surface melt on the AIS from passive microwave radiometry, previous studies have used a variety of statistical thresholding techniques that detect spikes in brightness temperature above a predetermined threshold, which are interpreted as melt events (Zwally and Fiegles, 1994; Torinesi et al., 2003; Picard et al., 2007). Typically, the threshold remains temporally constant over a year or longer and is based on mean brightness temperature for the 18 to 19 GHz frequency in the horizontal polarization. If the brightness temperature exceeds this threshold on a given day, then that day is assigned to be a melt day. However, variations in brightness temperature can also occur due to changes in snow density, physical temperature, and microstructure (Hofer and Mätzler, 1980; Mote and Anderson, 1995).

Fahnestock et al. (2002) and Johnson et al. (2020) partially circumvent this issue by checking for a bimodal distribution of brightness temperatures annually to determine the presence of melt. This technique can remove the effect of variations in annual wintertime temperature but does not account for day-to-day variation in temperature and snow stratigraphy. Tedesco (2009) developed a physics-based threshold for melt detection using the Microwave Emission Model for Layered Snow (MEMLS), but the threshold again only varies once per year. Therefore, current techniques do not fully and directly consider day-to-day variations in these physical properties of the snow. In part, this omission is because these variations in the properties of snow over the AIS are not well known. However, recent improvements in atmospheric reanalysis, firn modeling, and snow radiative transfer modeling allow us to capture these variations using a hybridized, physics-based technique for detecting melt with a dynamic threshold that varies day-to-day.

In this study, we address the need for a microwave melt detection technique that has a physical basis in the complex dynamics of snow microwave radiation. To achieve this, we create a dynamic threshold that captures the microwave effects of variability in snow properties from day to day, instead of a statistical threshold that only varies year to year. Our new, dynamic thresholding technique identifies melt days while quantifying the effects of snow temperature and density on brightness temperature. First, we use inverse radiative transfer modelling to determine grain size using the Community Firn Model (CFM) and the Snow Microwave Radiative Transfer model (SMRT) in non-melt conditions. This inversion step is similar to that of Mote and Anderson (1995), although we only apply this inversion for grain size and not density, as density is simulated by the CFM. Then, we use the outputs of CFM and SMRT to compute daily-varying thresholds. When a daily threshold is exceeded by the observed brightness temperature from the Advanced Microwave Scanning Radiometer (AMSR-2), we mark it as a melt day. We analyze our melt detection results from our hybrid, physics-based technique by comparing our results to Automatic Weather Station (AWS) data as well as results from statistical thresholding techniques. At thirteen AWS and the Larsen C, we validate our melt detection technique against observations and statistical thresholding techniques, as well as analyze our intermediary calculations of snow grain size during our melt detection technique.

**2 Data & Models**

## 2.1 Automatic Weather Stations

We run our melt detection technique for thirteen AWS to calculate both melt days and snow grain size. Ten of these sites experience melt and the other three are dry all year round. The ten melt sites have hourly melt rates from AWS-derived surface energy balance (SEB) analyses based on observed conditions (Jakobs et al., 2020). The hourly melt rates are converted to daily melt rates for comparison to microwave data. The three dry sites are Dome C (DC; 75.10ºS, 123.35ºE), Point Barnola (PB; 75.70ºS, 123.25ºE), and Kohnen (75.00ºS, 0.07ºE). We include these three dry snow sites (Dome C,

Point Barnola, and Kohnen) in our analysis of microwave grain size calculations and one dry snow site (Dome C) to validate the process of our melt detection technique.

The locations of all sites are shown in the map in Figure 1. These sites are found on grounded ice sheet, ice shelves, or ice rises as shown in Figure 1 (Jakobs et al., 2020). Our time series spans from July 2, 2012 to May 31, 2019 for all sites.

However, AWS-derived observations provide differing coverage of this time series depending on AWS location (Jakobs et al., 2020). Although we run the Hybrid Method for all thirteen sites, there is no crossover in time of AWS-derived data for AWS 4 and 6 with the AMSR-2 record. To note, the melt detection and microwave grain size calculations are independent of AWS observations so we are able to detect melt and calculate microwave grain size at all sites. However, we chose locations to maximize where AWS observations are available so that we can ultimately validate our melt detection at eight AWS

locations (AWS 18, AWS 17, AWS 14, AWS 15, AWS 19, and AWS 5).

## 2.2 Advanced Microwave Scanning Radiometer 2

AMSR-2 is a microwave radiometer onboard Japanese satellite Global Change Observation Mission – Water Satellite 1 (GCOM-W1).  This instrument provides 16 channels between 6.9 and 89 GHz. We use the 12.5 km x 12.5 km daily gridded product of AMSR-2 (Meier et al., 2018). We use the 18.7 GHz channel, as this frequency is most sensitive to the presence of

liquid water on the surface of the AIS. We use 18.7 GHz in the horizontal polarization (19H) for melt detection and 18.7 GHz in the vertical polarization (19V) for grain size. Penetration depth of this frequency varies widely between studies. Tikhonov et al. (2019) showed that the penetration depth of 19 GHz was a few tens of centimeters using the Special Sensor Microwave/Imager (SSM/I) passive microwave radiometer. Colliander et al. (2022) found the penetration depth at 19 GHz to be approximately 40 cm. The maximum depth of detection for this frequency was shown to be 1 to 2 m in Picard et al.

(2022a).

## 2.3 Community Firn Model

The CFM is an open-source model framework that simulates firn property evolution, including densification, temperature, melt, and grain growth (Stevens et al., 2020). Because surface grain size over the AIS is not well constrained, the grain

growth model for dry snow within CFM assumes a fresh snow grain radius of 0.1 mm across the AIS (Stevens et al., 2020).

When compared to surface grain sizes derived from the Moderate-Resolution Imaging Spectroradiometer (MODIS) Mosaic of Antarctica (MOA), this is an underestimate for many areas of the AIS associated with frequent melt events, especially ice shelves (Scambos et al., 2007; Haran et al., 2018). Therefore, we do not use grain size information from the CFM in our melt detection algorithm.

We force the CFM with daily Modern-Era Retrospective analysis for Research and Applications, Version 2 (MERRA-2)

precipitation and skin temperature (Gelaro et al., 2017). The CFM output is on the same spatial resolution as MERRA-2: 0.5º latitude by 0.625º longitude (Gelaro et al., 2017). The vertical resolution of the daily output profiles from the CFM varies spatiotemporally and is on the order of millimeters, which is too fine for input into the radiative transfer model. Radiative transfer models become inaccurate when layers are too small comparative to the wavelength used. Therefore, we merge thin layers from the CFM output to create thicker layers that are appropriate for the 18.7 GHz frequency. The layer merging

process is performed based on layer thicknesses of neighboring layers to fit the following general structure. From the surface down to 1 m, each layer of the CFM output is approximately 1 cm in thickness. From 1 m to 5 m, each layer is approximately 10 cm. A single, semi-infinite bottom layer spans 5 m to 50 m. The exact size of each layer varies slightly from day to day and location to location due to layer formation and merging within the CFM. The CFM output we use consists of snow temperature and density with depth and has daily temporal resolution.

**2.3 Snow Microwave Radiative Transfer model framework**

The Snow Microwave Radiative Transfer model framework, introduced in Picard et al. (2018), simulates brightness temperature and backscatter intensity while allowing for the use of several previously developed microstructure and scattering models, including the Dense Media Radiative Transfer model (DMRT; Tsang et al., 1985) and the Improved Born Approximation (IBA; Mätzler, 1998). These two models are shown to produce nearly equivalent results (Löwe and Picard,

2015). We chose to use IBA within SMRT, or hereafter SMRT-IBA.

Within SMRT-IBA, snow is represented as a two-phase medium (assuming no liquid water), with ice within a medium of air being the standard configuration. However, this approximation becomes less accurate at densities greater than 450 kg m$^{-3}$. Therefore, we switch to the opposite configuration of air within an ice medium when ice volume fraction exceeds 0.5. We achieve this by using the "dense snow correction" within SMRT-IBA. This allows us to model the emission and scattering of

snow at these higher densities, following the IBA example in Picard et al. (2022b). We considered an alternative model from Picard et al. (2022b) that uses the more advanced strong contrast expansion theory, but the model was not always numerically stable.

## 2.3 Microwave grain size

The snow microstructure model within SMRT-IBA that we chose to implement is the exponential model, which considers
grain size as a single parameter (Picard et al., 2018). Given that we are using an exponential model, the length scale input for
snow microstructure is referred to as the exponential correlation length ($L_C$). In this paper, we refer to this value as
"microwave grain size" ($L_{MW}$), a unifying concept that allows for simple comparison to other snow microstructure
parameters (Picard et al., 2022c). Microwave grain size ($L_{MW}$) is proportional to the Porod length ($L_P$), which relates snow
density ($\rho$), polydispersity ($K$), and optical grain diameter ($d_{opt}$) (Mätzler 2002; Picard et al., 2022c):

$$L_{MW} = L_C = K\,L_P = K\,(2/3)\,(1 - \rho/\rho_{ice})\,d_{opt} \tag{1}$$

Equation (1) shows that (a) microwave grain size is proportional to grain diameter, (b) larger optical grain diameters are
associated with higher microwave grain sizes, and (c) higher densities are associated with lower microwave grain sizes.
Therefore, microwave grain size is a complicated snow microstructural input into SMRT-IBA that relates to both grain
diameter and density.

We compare our calculations of microwave grain size within SMRT to in situ observation of specific surface area from two
snow pits in Picard et al. (2014). These data can be converted to microwave grain size given information about snow density
and polydispersity (Picard et al. 2022c). We also compare our results to optical grain radius as derived from the MODIS
MOA in 2013 (Scambos et al., 2007; Haran et al., 2018), a mosaic of MODIS imagery.

## 3 Methodology

We use the radiative transfer model SMRT to predict brightness temperature based on each day's modeled snow conditions.
Snow temperature, density, microwave grain size, and liquid water volume all affect snow brightness temperature and are
inputs into SMRT. We always set the snow's liquid water volume to zero regardless of how much liquid water is truly
present that day. In doing so, we compute dry snow brightness temperatures, which are lower than the brightness
temperatures that occur when snow is wet. In theory, our modeled dry snow brightness temperatures should equal observed
brightness temperatures on days with no liquid water present. We consider melt to have occurred on a given day when the
observed brightness temperature from AMSR-2 exceeds the modelled dry snow brightness temperature by at least a variable
amount related to dry snow microwave grain size (5 to 10 K).

## 3.1 Inputs into SMRT: Density, temperature, and microwave grain size

We input snow density and temperature from the CFM into SMRT. Additionally, snow microstructural parameters are also required for snow radiative transfer modeling, specifically microwave grain size for SMRT. However, accurate, daily-varying microwave grain sizes are not currently available across the AIS either from modelling or remote sensing. The CFM produces vertical grain size profiles, which can be converted to microwave grain size following Mätzler (2002), which is approximated for the Alps. A closer approximation can be made using Picard et al. (2022c). However, these profiles from CFM are highly dependent on fresh snow grain size, which is not currently well enough known across the AIS for use in microwave studies. Instead, we first use snow brightness temperature to determine snow microstructure, specifically microwave grain size, on days that we are confident that the snow is dry.

We can make this determination because brightness temperature for 18.7 GHz in both the horizontal and vertical polarizations decreases considerably as a function of microwave grain size (Figure 2a). Since brightness temperature is strongly influenced by microwave grain size, we solve for microwave grain size on non-melt days when microwave grain size is the only SMRT input variable that is unknown.

Brightness temperature also varies relative to temperature and density (Figures 2b and 2c). We note that there is a discontinuity in the relationship between brightness temperature and density (Figure 2c). This discontinuity is related to the challenge introduced by modeling snow at higher densities ($>450$ kg m$^{-3}$). As discussed in Section 2.3, the model switches from its standard "ice in air" configuration to "air in ice", resulting in a discontinuity when density is approximately 450 kg m$^{-3}$. Since Figures 2a to 2c are all idealized cases with uniform snow temperature and density with depth, this discontinuity is a discrete jump. However, in a real snow profile, snow temperature and density vary with depth. We show the dependence of microwave grain size, temperature, and density perturbations on brightness temperature from sample output from the CFM for Dome C in Figures 4d to 4f. The effect of model switch is integrated in Figure 2f but still represents a source of uncertainty introduced by radiative transfer modelling.

### 3.2 Polarizations used for melt detection and microwave grain size

In this study, we focus our melt detection analysis on the horizontal polarization as it has been shown to be sensitive to the presence of liquid water on the surface of the AIS, making it useful for melt detection (Zwally and Fiegles, 1994 and Torinesi et al., 2003). However, the vertical polarization can also be useful specifically in microwave radiative transfer modelling as it is generally easier to model the vertical component of emissivity than the horizontal component. This is because the vertical component of emissivity is not as sensitive to the detailed density structure of icy layers (Comiso et al., 1997; Durand et al., 2008). Therefore, we derive microwave grain size and use it to perform melt detection with the Hybrid

Method using 19H as this channel is more sensitive to melt (Section 3.3 to 3.4). We re-derive microwave grain size using
19V (Section 3.3) and consider this version to be our official microwave grain size product.

## 3.3 Calculating microwave grain size ($L_{MW}$)

To calculate microwave grain size, we must hybridize our method to a statistical thresholding technique presented in Picard
et al. (2022d). In this statistical thresholding technique, the mean brightness temperature from June to September within an
AMSR-2 grid cell acts as the "dry brightness temperature". A melt day is defined to occur when the daily brightness
temperature exceeds this dry brightness temperature by at least 20 K. This is a relatively low threshold, meaning that this
technique produces many melt days. We also compare our end results of the Hybrid Method to the statistical thresholding
technique introduced in Torenisi et al. (2003). Torenisi et al. (2003) describe a recursive method that involves first
calculating the annual mean brightness temperature plus $N$ times its standard deviation (we used $N = 3$). Then, they
recursively remove days that exceed this threshold, recalculating the mean brightness temperature and standard deviation
each time. The third method for comparison is that of Zwally and Fiegles (1994), where the threshold for melt is taken to be
30 K over the mean brightness temperature within an AMSR-2 grid cell for the time series. Note that all three of these
statistical thresholding techniques use the horizontal polarization.

We incorporate the statistical thresholding technique described in Picard et al. (2022d) to determine on which days we have
the ability to calculate microwave grain size based on brightness temperature observed by AMSR-2. Note that we have not
assigned melt days as defined by the Hybrid Method. Instead, we identify "potential melt days" as any day within +/- 7 days
of a melt day detected by the thresholding technique described in Picard et al. (2022d). We chose this value because 80% of
AWS-derived melt falls within this +/- 7 day window for the AWS used in this study. "Potential non-melt days" are outside
this +/- 7 day window. We chose the threshold from Picard et al. (2022d) over other statistical techniques because it uses a
threshold of only 20 K above the mean, horizontally polarized brightness temperature of its assigned dry season, typically
making it most sensitive to melt resulting in more "potential melt days" than other statistical thresholding techniques.

SMRT predicts brightness temperature based on input snow conditions. We employ what is termed as "inverse radiative
transfer modeling". This technique was used in Picard et al. (2012) to calculate grain size for Dome C. Unlike Picard et al.
(2012), we use a lower, single frequency (18.7 GHz) and assume vertically homogenous grain size. One way to employ
inverse radiative transfer modeling is to run the radiative transfer model for a range of monotonically increasing input
microwave grain sizes and then select the microwave grain size that produces a modeled brightness temperature closest to
the observed brightness temperature. In our methodology, we apply a slightly different process that yields equivalent results
but is optimized for computational efficiency.

For each potential non-melt day, we run SMRT once with an overestimate or a "high" value for microwave grain size ($L_{initial}$ + *bound*) and a second time with an underestimate or a "low" ($L_{initial}$ - *bound*) value for microwave grain size. The values for

$L_{initial}$ and *bound* used are denoted in the table in Figure 3b*. We assume temperature and density profiles based on CFM outputs. This results in both an overestimate and an underestimate for brightness temperature, as brightness temperature decreases as a function of microwave grain size at this frequency. Using these two endmembers as bounds, we linearly interpolate to the observed brightness temperature to find a microwave grain size associated with this value. We repeat this process several times over with more accurate microwave grain sizes with narrower bounds, according to the table in Figure

3b*. Eventually, we converge on a microwave grain size for that day that, when input into SMRT, produces a modeled brightness temperature that is approximately equal to the observed brightness temperature. These steps are shown in Figure 3a to 3e. This process based off the Secant method (Wolfe, 1959). This minimization results in modeled microwave grain sizes during potential non-melt days that, when input into SMRT along with temperature and density from CFM, each produce a brightness temperature that is within 0.1 K of its respective AMSR-2 observation.


To make this process more computationally efficient, we break this step into three separate sub-steps, first running the Secant method less frequently in time to generate more accurate initial microwave grain size ($L_{initial}$) so we can converge on AMSR-2 brightness temperatures faster when we run it at a daily resolution. First, we estimate a general microwave grain size for that location for Sub-step #1: Single $L_{MW}$ by using any single potential melt day for that location. Then, we use the

output $L$ from Sub-step #1 to solve for microwave grain sizes for the first day of each month in Sub-step #2: Monthly $L_{MW}$. Finally, in Sub-step #3: Daily $L_{MW}$, using the nearest $L$ in time from Sub-step #2 as initial input, we cycle converge on daily microwave grain sizes that, when input into SMRT along with output from CFM, best matches the local observed brightness temperature.


After we have established microwave grain sizes for each of the potential non-melt days, we consider microwave grain size on potential melt days. Since melt dramatically increases brightness temperature, we cannot compute microwave grain size during potential melt days using the method described above. In the absence of an accurate microwave grain size dataset or model output, we determine the microwave grain size on potential melt days by linearly interpolating between the non-melt

days (Figure 3f). This results in our final microwave grain sizes, $L_{MW}$, every day across the time series.

### 3.4 Assigning melt days and non-melt days

Now that we have a microwave grain size product, we can calculate brightness temperatures under the assumption that every

day is dry. To do this, we use the temperature and density profiles from CFM for each day as inputs for SMRT. This generates dry snow brightness temperatures, $T_{B,dry\ snow}$, as seen in Figure 3g. To calculate our threshold brightness

temperature, we subtract a value from microwave grain size. Note that we specifically subtract from microwave grain size to find a dynamic threshold that is always higher than the dry snow brightness temperature, as brightness temperature decreases with microwave grain size.


For this subtraction, we use the four-fold mean of a 31-day windowed running standard deviation of microwave grain size along our time series for potential non-melt days (hereafter referred to as $4\overline{\sigma_{LMW}}$). We chose $4\overline{\sigma_{LMW}}$ because our temporal linear interpolation across the potential melt days would yield a poorer estimation of microwave grain size for locations where microwave grain size is varying significantly from day to day. As shown in Figure 3g, we input this lower bound on

microwave grain size into SMRT along with the CFM temperature and density information to generate a dynamic threshold, $T_{B,threshold}$, that varies from day to day. The presence of liquid water causes an increase in brightness temperature; we interpret any day where the AMSR-2 brightness temperature exceeds this dynamic threshold as a melt day and any day that falls below as a non-melt day. This is shown in Figure 3h. We call this the Hybrid Method.

**3.5 Validation of dry snow zone**

The purpose of this study is to detect the presence of liquid water in a physically based way and assess the technique's usefulness in detecting melt. Before applying our dynamic thresholding method to determine melt days, we test how AMSR-2, SMRT, and CFM can be used together to estimate brightness temperature at a dry site. We use Dome C, a cold site on the

Antarctic Plateau, as our test site. Figure 4a shows that the brightness temperature in 19H never exceeds the Picard et al. (2022d) threshold, so, by default, 0% of the time series would be considered to have melt according to the Hybrid Method as we detect no potential melt days.

As seen in Figure 4b, the microwave grain size has a mean of 0.18 mm for 19V and appears to have a seasonal signal; higher

microwave grain sizes occur during the austral winter, and lower microwave grain sizes occur during the austral summer. The amplitude of this oscillation is about 0.01 mm, or 5% of the mean microwave grain size for this site. Since the CFM-modeled snow and firn temperature is influenced by MERRA-2 skin-temperature forcing, we compare the MERRA-2 skin temperature to in situ temperature data from Dome C to check for a seasonal bias. Indeed, there is a seasonal bias in the MERRA-2 skin temperature for this site, and we remove it using a linear regression. The relationship between the corrected

temperature ($T_{corrected}$) and air temperature in CFM is: $T_{corrected} = T_{CFM} * 1.066 – 9.246$. We re-run the CFM with the bias-corrected (BC) temperature input and we recalculate microwave grain size using SMRT. Figure 4b shows that the microwave grain size no longer varies seasonally after our bias correction. The microwave grain size also increased in the BC run by a mean of about 10%.

We compare our SMRT-derived microwave grain sizes to two snow profiles located at Dome C that are approximately 7 m in depth (Picard et al., 2014). We would not expect an exact match for this comparison due to issues with representativeness both (a) spatially, within the 12.5 x 12.5 km AMSR-2 grid cell, and (b) with depth, as penetration depth of AMSR-2 at 19 GHz may be higher or lower than 7 m, as noted in the Section 2.2. The in situ- and SMRT-derived microwave grain sizes fall between 0.16 mm and 0.21 mm (Figure 4b).


Our results show that neither the dry snow brightness temperatures nor our dynamic threshold are significantly affected by the bias correction. Though the bias correction does result in about a 15% change in microwave grain size for Dome C, this correction does not significantly impact the dry snow brightness temperature nor our dynamic threshold. This is by design, as dry snow brightness temperature is forced to AMSR-2 during all potential non-melt days. In Figure 4c, the difference is not

visible as they plot exactly on top of each other. The bias corrected threshold ($-4\overline{\sigma_{LMW}}$) also plots exactly on top of the non-bias threshold in Figure 4c. Since the errors into our dynamic threshold are small, we are satisfied with using the non-BC temperature forcing. However, we acknowledge that there may be biases in microwave grain size as all errors in temperature, density, and radiative transfer modeling are propagated there.

**4. Results**

**4.1 Melt detection**

**4.1.1 Automatic weather stations**


Now that we have established the feasibility of using SMRT and CFM to model Antarctic brightness temperatures, we branch out to sites that experience seasonal melt. We first focus on one site: AWS 18 on the Larsen C ice shelf. For AWS 18, there are days where AMSR-2 brightness temperature exceeds the thresholds from Zwally and Fiegles (1994), Torinesi et al. (2003), and Picard et al. (2022d) each year, seen in Figure 5a. The Hybrid Method can detect extra days of melting occurring

within a 15-day window of a particular melt day identified by Picard et al. (2022d) due to our +/- 7 day window for potential melt days. Given the numerous occasions where AMSR-2 brightness temperatures surpass the Picard et al. (2022d) threshold each year, there are a substantial number of days that the Hybrid Method could classify as melt. To quantify this number of days, the first step is computing microwave grain size at this site.

The mean microwave grain size for AWS 18 is 0.36 mm, shown in Figure 5b. The linearly interpolated microwave grain size increases from the austral spring to the austral fall for each of the four austral summers shown, suggesting grain growth due to wet snow metamorphosis during the austral summer (Brun, 1989). In Figure 5c, the dry snow brightness temperature lines up exactly with the AMSR-2 brightness temperature because of how we calculated the microwave grain size during all

potential non-melt days. The propagated $4\overline{\sigma_{LMW}}$ thresholding bound we use to determine melt days varies from about 170 K

to 210 K across this time series. Melt days make up 29.6% of the time series using the Hybrid Method and 27.1% of the time series according to AWS-derived melt data. In Figure 5d and 5e, melt days and non-melt days that are assigned by the Hybrid Method match with AWS-derived melt rates at a rate of 89.2% of the time series, which is second best of the techniques we studied. The Picard et al. (2022d) thresholding technique matches SEB-derived melt rates best for this location at 89.3% (Table 1). For this AWS, there are a similar number of melt days assigned by the Hybrid Method and

Picard et al. (2022d), and fewer melt days assigned by Torenisi et al. (2003) and Zwally & Fiegles (1994).

Shown in Figure 6a, Picard et al. (2022d) and the Hybrid Method at AWS 17 match with AWS-derived melt rates 91.0% and 91.2% of the time series, respectfully. Additionally, for all AWS shown besides AWS 19, there are isolated melt events near or during the austral summer that are detected by AWS-derived data that are not identified in any of the melt detection

techniques, shown in Figure 6a, 6b, 6c, and 6e. The technique that detects melt the best varies from site to site, with each technique matching best with AWS for at least one site, according to Table 1. AWS 11 and 16 are not shown in Figure 6 as they both had very infrequent melt and none of the melt detection techniques identified any melt days at these locations, shown in Table 1. Overall, the technique that matched best was Torenisi et al. (2003) matching with AWS at a rate of 94.6%, weighted by the number of days in each site. However, all sites fell between 94% and 95% matching to AWS-derived data;

all techniques had similar overall performance compared to AWS-derived data.

### 4.1.2 Larsen C ice shelf

For the 2013-2014 melt season, we applied the Hybrid Method and compared it to the statistical thresholding techniques,

shown in Figure 7. The spatial mean melt duration, or number of melt days across a melt season, for the Hybrid Method was 61 melt days, which was lower than Picard et al. (2022d) by five melt days. However, the average number of melt days was higher than Torenisi et al. (2003) and Zwally and Fiegles (1994), which were 44 and 36, respectively. The region with the biggest difference between our Hybrid Method and Picard et al. (2022d) method is outlined in the southern portion of the Larsen C in a black box in Figure 7d.


Melt onset for the Hybrid Method and Picard et al. (2022d) is similar (Figure 7b and 7e). Average melt onset is 12 and 27 days later for Torinesi et al. (2003; Figure 7h) and Zwally and Fiegles (1994; Figure 7k) than the Hybrid Method, respectively. The pattern of melt onset is similar for all four methods, with earlier melt onset closer to the grounding line (Figure 7c, f, i, and l). The similarity in spatial pattern of between melt onset does not hold for melt end date amongst each

melt detection technique. While all four methods generally have later melt end dates near the grounding line and earlier melt dates closer to the periphery of the ice shelf, the Picard et al. (2022d) method has a pattern of later melt end dates in the southern interior portion of the ice sheet shown by the boxed outline.

Luckily, AWS 14 and AWS 15 both fall in this area so we can compare these methods to AWS-derived data. For both sites, the Hybrid Method identifies fewer melt days than Picard et al. (2022d), which is closer to that of AWS-derived data, shown in Figures 8a and 8c. In Figure 8a, for AWS 14, Picard et al. (2022d) indicate four additional melt days that fall from March to May even though the final melt day according to AWS-derived data is February. For AWS 14 and 15, in Figures 8b and 8d, the additional days marked as melt by Picard et al. (2022d) fall in the beginning of the melt season. In Figures 8b and 8d, these extra sites identified by Picard et al. (2022d) fall between our dynamic threshold and our dry snow brightness temperature, as expected by our methodology. It is notable that, in Figure 8a, the majority of AWS-derived melt ends at the end of January for AWS 14. However, seen in Figure 8b, AMSR-2 brightness temperatures remain elevated during first 8 days of AMSR-2 and thus is considered melt by both the Hybrid Method and Picard et al. (2022d). A similar pattern is noticeable a week earlier in for AWS 15 in Figures 8c and 8d.

## 4.2 Microwave grain size

### 4.2.1 Comparison to MOA

In addition to performing melt detection, the Hybrid Method also produces microwave grain size information. In Figure 9a, microwave grain size varies from location to location. Among these thirteen sites, lower microwave grain sizes are found on grounded ice sheet. Slightly higher microwave grain sizes are found on the ice shelves of Dronning Maud Land. Microwave grain sizes are highest at the Antarctic Peninsula, for the Larsen B ice shelf remnant site and the three Larsen C sites. We compare our microwave grain sizes to optical grain radius from the 2013 MOA, which has a similar pattern to microwave grain size across the thirteen AWS (Haran et al., 2018). The Pearson correlation coefficient between these two grain size variables is 0.88 ($p < 0.01$), as shown in Figure 9b.

The spatial pattern in optical grain radius across the Larsen C from the 2013 MOA is shown in Figure 10, at 750 m resolution in Figure 10a and upscaled to the AMSR-2 grid in Figure 10b. There is generally higher optical grain radius from 2013 MOA both southward and toward the grounding line. Microwave grain size decreases gradually across the Larsen C from lower latitudes to higher latitudes. Additionally, it has slightly lower microwave grain sizes closer to the grounding line. The Pearson correlation coefficient between optical grain radius and microwave grain size across the Larsen C is 0.34 ($p < 0.01$).

### 4.2.2 Comparison to melt duration

In addition to optical grain radius, microwave grain size also appears to have a relationship with melt duration. In Figure 11a, we can see lower values of mean microwave correlation length on the grounded ice sheet along with lower or zero melt duration. On Dronning Maud Land there are moderate durations of melt and moderate microwave grain sizes, and on the Antarctic Peninsula there are longer durations of melt and larger microwave grain sizes. In Figure 11b, there is a relationship between microwave grain size and melt duration with a Pearson correlation coefficient of 0.94 ($p < 0.01$) for these thirteen sites. For more further information on melt detection and microwave grain size, see the Supplemental Materials for the results from mid-2012 to mid-2019 that were used to create these temporal averages.

Across the Larsen C, we can see a somewhat similar spatial pattern in microwave grain size in Figure 12a to melt duration in Figure 12b for the 2013-2014 melt season. These two values are related at a Pearson correlation coefficient of 0.56. In Figure 12c, we consider "total threshold exceedance"; total threshold exceedance is the integrated exceedance of the AMSR-2 brightness temperatures above the Hybrid Method's dynamic threshold. Total threshold exceedance and microwave grain size are correlated at a Pearson correlation coefficient of 0.92 across the Larsen C.

## 5 Discussion

### 5.1 Intercomparison of melt detection techniques

In this study, we developed a melt detection algorithm that is robust to temperature and density variations. In certain scenarios, this technique can perform better than the technique to which it is hybridized. This is the case for AWS 14 and AWS 15 on the Larsen C ice shelf during the 2013-2014 melt season. However, when considering eight AWS across multiple melt seasons, we discovered that the Hybrid Method does not necessarily improve the agreement between melt events determined by AWS-derived melt data and predicted by the Hybrid Method.

The Hybrid Method does not appear to improve melt detection potentially because statistical thresholding techniques and physics-based techniques are both able to capitalize on the strong relationship between the presence of liquid water in snow and the brightness temperature of the 19H channel. All methods perform similarly against AWS-derived data at similar rates. Therefore, for the purposes of large-scale, ice sheet-wide melt detection where melt volume and grain size are not of interest, we still recommend the use of statistical thresholding technique due to their computational efficiency over our physics-based method. Given our comparison to AWS-derived data, out of the three statistical thresholding techniques tested, we recommend the Torenisi et al. (2003) method as it matched best with AWS-derived melt data. The Picard et al. (2022d) method performs second-best among the melt detection techniques. In the development of this method, their goal in melt detection was "simplicity and reproducibility" for eight sites rather than designing the most physical and robust algorithm for meltwater detection across the AIS (Picard et al., 2022d). The fact that it generally performs well shows that even techniques

designed for simplicity can have similar performance to more advanced techniques for melt detection. While these techniques perform relatively well against the AWS-derived data, the Hybrid Method is still especially important given its basis in physics. Future development, such as using 5-daily resolution for microwave grain size, could increase computational efficiency for the Hybrid Method and allow for its use across larger areas. Moreover, future studies could use the Hybrid Method to develop more robust statistical thresholding techniques that can be readily used across the AIS.

## 5.2 Limitations of the melt detection validation using AWS

A key finding is that all melt detection techniques performed relatively similarly in comparison to AWS-derived melt. There appears to be a limit to their performance due to several factors related to how we validate melt. One factor is that AWS-

derived melt data deduce the melt volume produced on a certain day, whereas a "melt day" indicates the presence of liquid water on the ice sheet as derived from daily microwave data. While we expect these two variables to be very well correlated, we do not expect them to match exactly because liquid water can persist on the surface of the ice sheet even though the snow is not actively melting. Furthermore, since melt may not always occur during overpasses, AMSR-2 may miss or preferentially detect melt. Differences can also result from the large footprint of the microwave radiometer. Finally, AWS-

derived melt data contain some uncertainty associated with both measurement error and model accuracy (Jakobs et al., 2019). For these reasons, it is challenging to validate melt detection from microwave radiometry using AWS-derived data, which is a similar finding to that of de Roda Husman (2022). It is possible that one of the melt detection methods is truly preforming better than the others, but that conclusion may be clouded by our comparison to AWS-derived data.

## 5.3 Limitations of the Hybrid Method

At present, the Hybrid Method is tied to a statistical thresholding technique described in Picard et al. (2022d); our method limits the potential days that could be assigned melt to a 15-day window around days that are considered melt by Picard et al. (2022d). While this allows our method to detect melt days that are clustered close to melt events detected by Picard et al.

(2022d), it also prevents our method from detecting any melt events that are isolated in time, such as over the austral winter. However, future work may allow for this method to be independent from statistical thresholding techniques. For example, future work could include making the potential melt days self-consistent with the calculated melt days in an iterative fashion. This method would then be able to identify some small or localized melt events that are isolated in time from larger melt events that are detected by statistical thresholding techniques.

Given that CFM can be run across the AIS, the Hybrid Method could be run for ice sheet-wide studies as well. However, a limitation of the Hybrid Method is its computational efficiency. Inverting SMRT to calculate microwave grain size takes several iterations and is highly computationally expensive. Running the Hybrid Method for the 2013-2014 melt season

across the Larsen C Ice Shelf took about five days run on a local machine. Running the Hybrid Method on an ice sheet-wide scale would require careful consideration of computational resources, and likely the use of supercomputing resources. A small change to somewhat improve speed would be implementing the Brent method instead of the Secant method during our inversion to calculate microwave grain size (Brent, 1973).

Other limitations of the Hybrid Method include uncertainty related to modeling brightness temperature for snow densities over 450 kg m$^{-3}$. A potential improvement could be made by using the Strong Contrast Expansion (SCE) instead of using a modifying version of IBA (Picard et al. 2022b). However, the melt detection technique forces dry snow brightness temperature to AMSR-2, which reduces error propagated from either firn modeling or radiative transfer modeling.

Additionally, we acknowledge that, though our dry snow brightness temperatures are physics-based, our fixed bound on microwave grain size ($-4\overline{\sigma_{L_{MW}}}$) with a 31-day window to calculate the dynamic threshold is based in statistics. We chose this bound for the melt threshold because higher variations in microwave grain size on a day-to-day basis indicate higher uncertainty in our linear interpolation of microwave grain size across the melt season, and thus increase uncertainty in our dry snow brightness temperatures. In theory, if our firn and radiative transfer modeling were perfect and we knew microwave grain size exactly, we could set this bound to zero and use dry snow brightness temperature as our threshold. While we compare the Hybrid Method to AWS-derived data, future work could instead use the AWS-derived data to better calibrate the Hybrid Method. Additionally, improvements in firn modeling, especially of grain size, would allow us to more accurately model dry snow brightness temperature and allow us to reduce this bound on microwave grain size by increasing the accuracy of dry snow brightness temperature.

**5.4 Applications & limitations of microwave grain size**

A key advantage of the Hybrid Method over the statistical thresholding techniques is that the Hybrid Method provides microwave grain size, an additional piece of information that may be useful in multiple ways. In our comparison of the Hybrid Method's microwave grain sizes to optical grain radius from MOA, we see a significant relationship between these two values. We do not expect a one-to-one correlation to MOA. This is primarily because the 18.7 GHz frequency is sensitive to the upper 5 to 10 m of snow, whereas MODIS is predominantly sensitive to the first 1 cm of snow (Lyapustin et al., 2009).

In deriving microwave grain size, we identify a positive correlation between microwave grain size and average melt duration. Any error in the CFM's temperature and density information could theoretically propagate into the following four parameters: microwave grain size, dry snow brightness temperature, our dynamic threshold, and number of melt days, as

illustrated by the flow chart in Figure 3. However, that alone would not necessarily lead to a positive correlation between the Hybrid Method's microwave grain size and the average melt duration as assigned by the Hybrid Method.

To illustrate the propagation of these biases, we describe an example scenario where MERRA-2 has biased-high skin temperature for a particular site. Then CFM would have a biased-high snow temperature profile. We solve for microwave grain sizes on non-melt days that, along with CFM density and biased-high temperature profiles, result in modeled dry snow brightness temperatures that match AMSR-2. Microwave grain size will be overestimated, contributing to a reduction in dry snow brightness temperature to compensate for the artificial increase caused by the biased-high temperature profile. Dry

snow brightness temperature on non-melt days is, by definition, robust to errors in SMRT and MERRA-2 because it is forced to match AMSR-2. However, overestimating microwave grain size which may mean also overestimating the standard deviation in microwave grain size, resulting in a biased-high dynamic threshold and fewer detected melt days. The reverse would be true for underestimating MERRA-2 skin temperature. SMRT input biases would act to reduce the correlation between microwave grain size and average melt duration.


Therefore, the relationship between microwave grain size and melt duration is likely physical in nature and suggests that the snow microstructure is related to the percentage of the year in which snow undergoes melt. This relationship has a physical basis; wet snow metamorphism leads to faster grain growth than dry snow metamorphism (Brun, 1989), which is likely why locations that experience significant melt tend to have higher microwave grain sizes. These variations in microwave grain

size and snow texture are important because they affect the thermal conductivity, permeability, dielectric properties, and optical properties of snow. Therefore, microwave grain size is highly relevant for ice sheet research, especially satellite altimetry, as snow microstructure influences radar penetration depth. However, there are limitations to the utility of the microwave grain size product of the Hybrid Method, as it currently stands. Our calculations of microwave grain size refer to a bulk layer with a depth that depends on penetration depth of AMSR-2 at that frequency. Indeed, the penetration depth at a

given frequency depends on microwave grain size. This circuitous relationship means that the representativeness of our microwave grain sizes to the physical snowpack is unclear. However, future developments such as using two different frequencies to establish an approximate surface microwave grain size and slow with depth could help alleviate this issue, in a similar way as Picard et al. (2012).

**5.5 Goal of melt quantification**

The ultimate reason for this study is to build on melt detection research with the goal of melt quantification across the AIS. Currently, observations of melt quantity in Antarctica are limited to a small number of in-situ surface energy balance sites. In this study, we show the feasibility of using a satellite- and physics- based approach to melt detection. With additional

refinement and calibration, this Hybrid Method has potential to estimate melt volumes instead of simply detecting the

presence of melt. In a similar way to how we used inverse radiative transfer modeling to calculate microwave grain size, we could potentially use it to solve for melt volume by essentially capitalizing on the degree to which AMSR-2 exceeds dry snow brightness temperature in a way that is informed by radiative transfer modeling. The strong correlation between total threshold exceedance and microwave grain size may hint that the total threshold exceedance may be related to the amount of

liquid water present because larger amounts of liquid water may increase wet snow metamorphism. However, it is worth noting that this increased correlation may also be related to the proportion of an AMSR-2 grid cell where liquid water is present and not a quantification of liquid water volume.

An algorithm that solves for even a rough estimate of melt volume would constitute a key step forward in understanding

surface melt across the AIS. This improvement would likely require the use of multiple frequencies and polarizations. This inverse problem is especially complex because liquid water has varying stratification vertically. More testing would need to be done to not only see what volume and vertical distribution of melt would saturate the microwave signal at a given frequency, but also understand the effects of localized melt within the 12.5 km AMSR-2 footprint. Quantifying surface melt from microwave radiometry is a complex problem; however, addressing it would greatly improve our understanding of

surface melt on ice sheets and ice shelves.

**6 Conclusions**

Using the Hybrid Method, we can estimate the microwave grain size of snow by combining firn-model outputs and inverse radiative transfer modeling forced by AMSR-2 for dry snow conditions. Next, we show how to interpolate the microwave

grain size across the melt season, using it along with information from CFM to forward model brightness temperature assuming dry snow conditions. This approach allows us to create a dynamic, hybrid threshold to identify melt days in a way that addresses day-to-day fluctuations in weather and snow conditions. Bias correction of reanalysis forcing can affect modeled microwave grain size but only minorly influences the dry snow brightness temperature. Examining eight sites that experience melt and comparing to AWS-derived melt data, we find that the Hybrid Method performs similarly to the

thresholding techniques. Running the Hybrid Method across the Larsen C ice shelf for the 2013-2014 melt season reveals an area of overestimate in melt days that appears to be reined in by the Hybrid Method. The Hybrid Method also has the advantage of providing microwave grain size, which is correlated with MOA grain radius and the Hybrid Method's melt duration. This work shows that we can combine firn modeling with radiative transfer modeling to determine properties of the snow, potentially expanding the use of snow radiative transfer modeling beyond point locations with observational data. This

methodology can be further developed for use in melt volume studies, as opposed to simply melt detection.

## Author Contributions

MD and BM designed the methods and analyzed them. CMS improved analysis and contributed to discussions and conclusions. MD prepared the manuscript with help from all other co-authors.

## Competing Interests

The authors declare that they have no conflict of interest.

## Data Availability

The dataset can be found on our Zenodo repository at https://doi.org/10.5281/zenodo.10529574.

## Acknowledgements

This work is funded by the Future Investigators of National Aeronautics and Space Administration Earth and Space Science
and Technology (FINESST), Grant no. 80NSSC20K1643. The work was also supported by the NASA Interdisciplinary Research in Earth Sciences and Cryospheric Sciences programs.

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

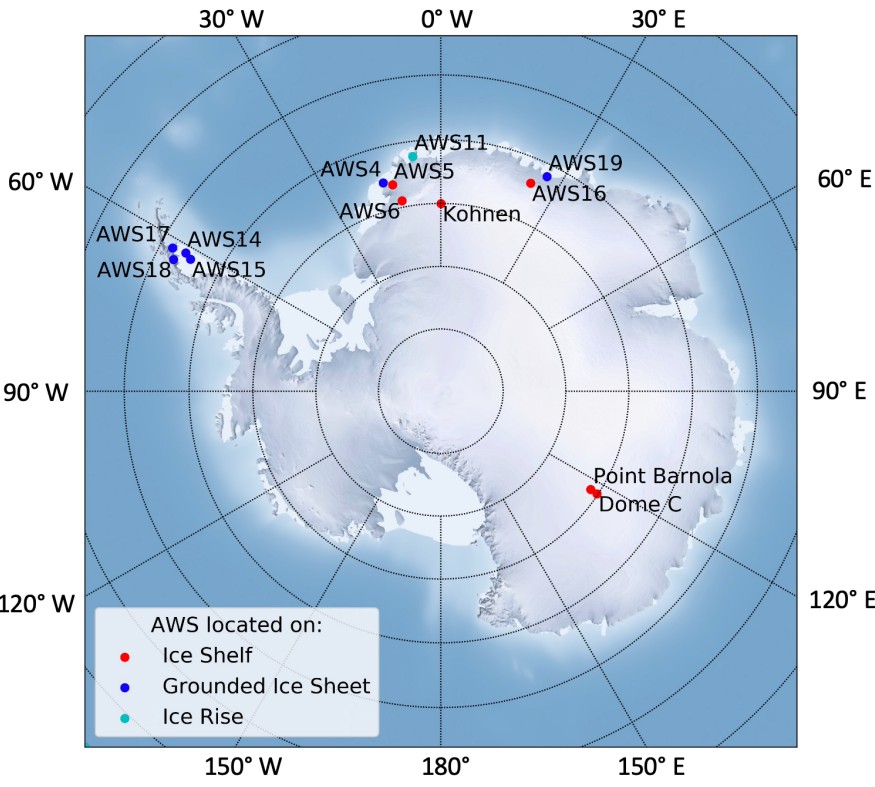


**Figure 1: A map of the locations of the thirteen AWS. Sites on grounded ice sheet are red, sites on ice shelves are blue, and sites on ice rises are cyan. Background image from www.shadedrelief.com.**


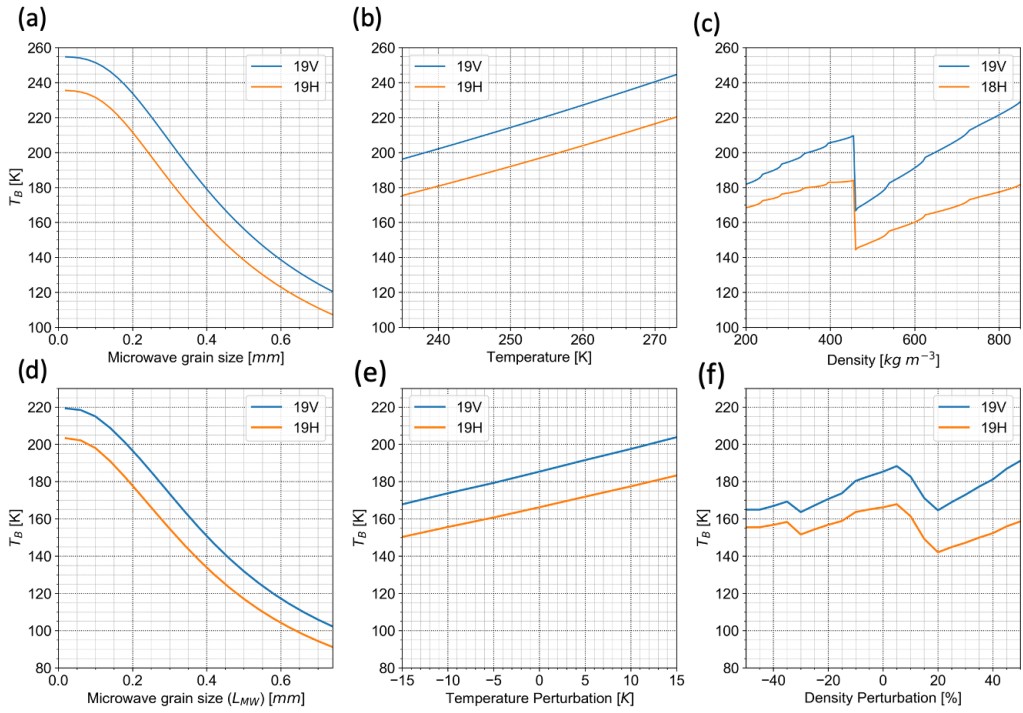

**Figure 2: Brightness temperature varies based on microwave grain size, temperature, and density input into SMRT.**
**In the three upper subplots (a-c), we vary one input into SMRT while holding the other two input variables constant**
**for an idealized, uniform snowpack. In these three lower subplots (d-f), we vary one input SMRT variable relative to**
**depth-varying, realistic snow profile (CFM snow profile for January 1st, 2014 at Dome C). (a) Holding temperature**
**and density constant (at 265 K and 375 kg m-3), the blue and orange lines show how brightness temperature varies**
**with respect to microwave grain size for 19V and 19H, respectively. (b) Holding microwave grain size and density**
**constant (at 0.25 mm and 375 kg m-3), the blue and orange lines show how brightness temperature varies with respect**
**to temperature for 19V and 19H, respectively. (c) Holding microwave grain size and temperature constant (at 0.25**
**mm and 265 K), the blue and orange lines show how brightness temperature varies with respect to density for 19V**
**and 19H, respectively. (d) Forcing density and temperature to the realistic snow profile, the blue and orange line**
**represent brightness temperature as we vary microwave grain size for 19V and 19H, respectively. (e) Holding**
**microwave grain size constant at 0.25 mm and forcing density from the realistic CFM profile, the blue and orange**
**lines represent brightness temperature as we perturb temperature by a given value constant with depth to the**
**example CFM profile for 19V and 19H. (f) Holding microwave grain size constant at 0.25 mm and forcing**
**temperature from the example CFM profile, we perturb density by the density perturbation percentage multiplied by**
**the air volume fraction, for 19V and 19H, respectively.**

**STEP 1: Calculate microwave grain size ($L_{MW}$) for dry days**

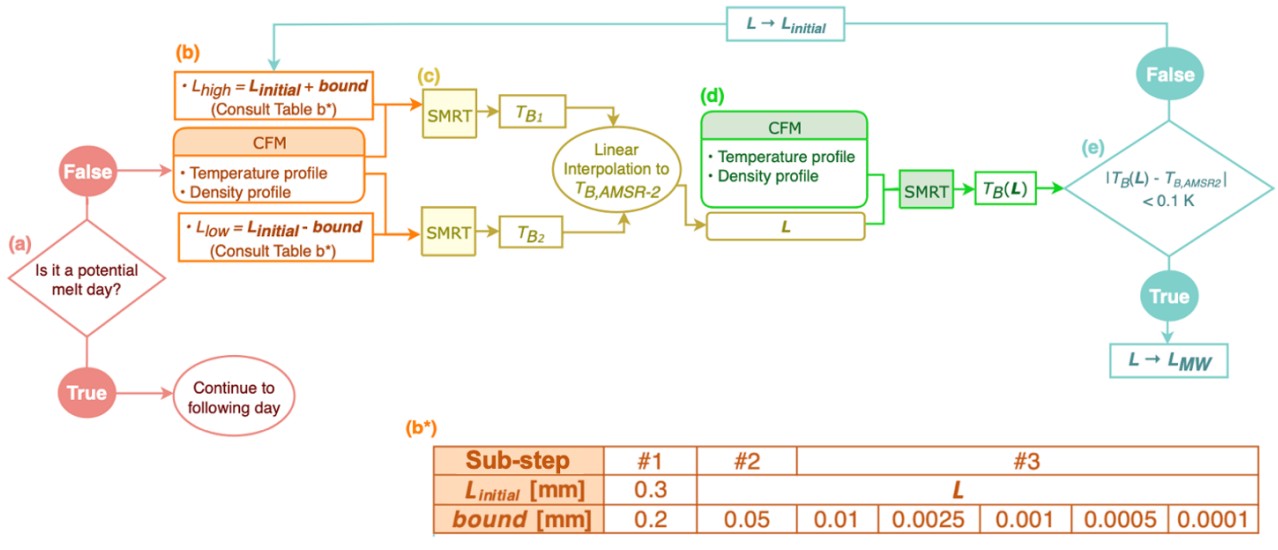

| Sub-step | #1 | #2 | #3 | | | | | |
|---|---|---|---|---|---|---|---|---|
| $L_{initial}$ [mm] | 0.3 | | $L$ | | | | | |
| **bound** [mm] | 0.2 | 0.05 | 0.01 | 0.0025 | 0.001 | 0.0005 | 0.0001 | |

**STEP 2: Assign Melt days/Dry Days**

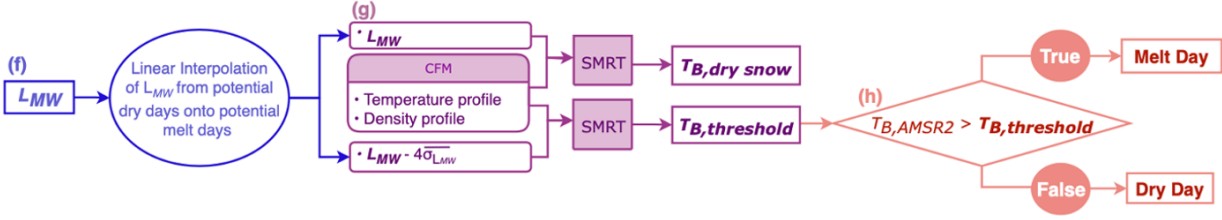

**Figure 3: Flow chart showing the hybrid-method algorithm used to calculate melt days. Step 1 (top) illustrates how microwave grain size ($L_{MW}$) is calculated using CFM and SMRT using repeated linear interpolations. Step 2 (bottom) illustrates how brightness temperature ($T_{B, dry\ snow}$) and its dynamic threshold ($T_{B,threshold}$) can then be calculated from $L_{MW}$, and used to determine melt days versus non-melt days. (a-h) are individual steps described in the main text.**

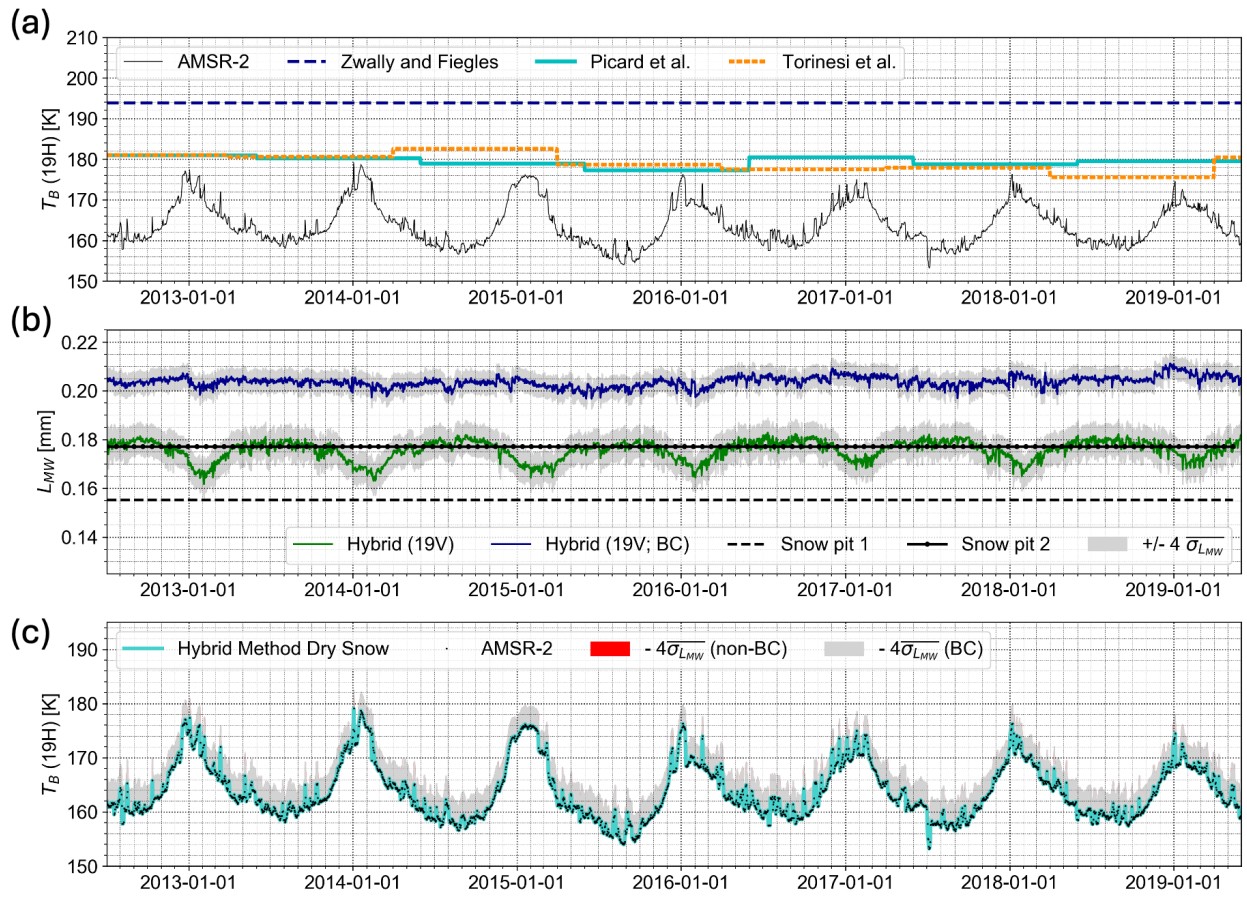

**Figure 4: A seven-year time series that illustrates the Hybrid Method's process of generating a dynamic threshold for melt detection at a dry site, Dome C. (a) Brightness temperature from AMSR-2 (19H; black line) with thresholding techniques from Zwally and Fiegles (1994; dashed dark blue line), Torinesi et al. (2003; dotted orange line), and Picard et al. (2022d; solid cyan line). (b) Microwave grain size from two Dome C snow pits (Picard et al., 2014). Microwave grain size computed from the Hybrid Method (19V; green line) and the Hybrid Method bias corrected version (19V; navy line) along with grey bounds that represent $4\overline{\sigma_{L_{MW}}}$. (c) AMSR-2 brightness temperature (19V; black crosses). Hybrid Method dry snow brightness temperature, bias corrected version (19V; cyan line) along with propagated $4\overline{\sigma_{L_{MW}}}$ (grey area). This is plotted on top of dry snow brightness temperature non-bias corrected version and the propagated $4\overline{\sigma_{L_{MW}}}$ (red area). The propagated non-bias corrected (red area) and bias corrected versions (grey area) are almost identical, causing the grey area to eclipse the red area in (c).**



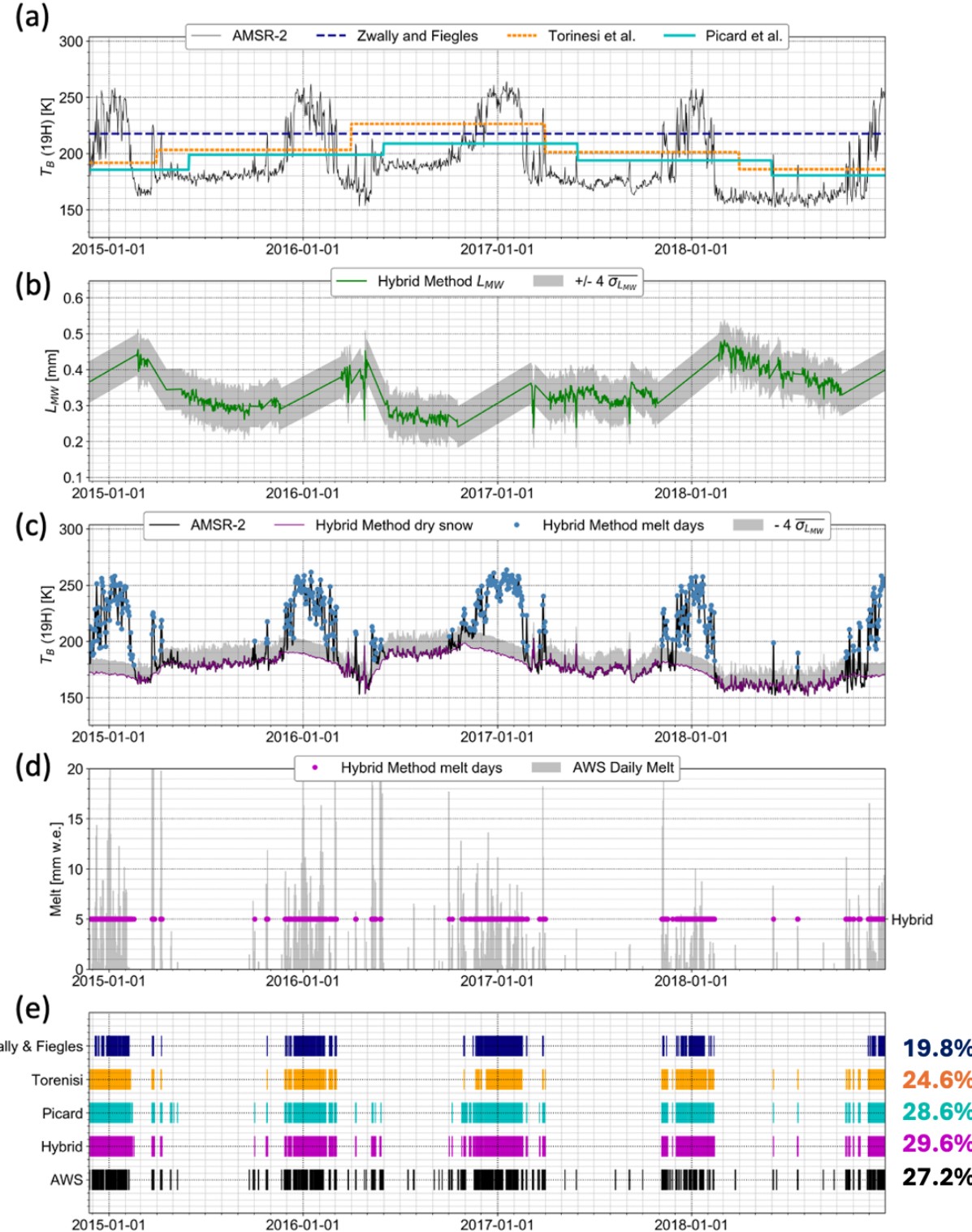

**Figure 5: Calculation of microwave grain size and detection of melt days using the Hybrid Method compared to AWS 18 observations and statistical thresholding techniques. (a) Brightness temperature from 19H from AMSR-2 (black line) with thresholding techniques from Zwally and Fiegles (1994; dashed dark blue line), Torinesi et al. (2003; dotted orange line), and Picard et al. (2022d; solid cyan line). (b) Microwave grain size computed from the Hybrid Method (19H; green line) along with +/- 4$\overline{\sigma_{L_{MW}}}$ bounds. (c) AMSR-2 brightness temperature (19H; black line). Hybrid Method dry snow brightness temperature (19H; purple line) with propagated -4$\overline{\sigma_{L_{MW}}}$ (grey bounds). Melt days detected by Hybrid Method (blue dots). (d) Melt days detected by Hybrid Method (purple dots). These represent melt days versus non-melt days in a binary fashion and do not reflect a specific melt volume. AWS surface energy balance melt data (grey bars). (e) Days when AWS surface energy balance melt > 0 (black lines). Melt days detected by Hybrid Method (purple lines). Melt days detected by Picard et al. (2022d; cyan lines), Torinesi et al., (2003; orange lines), and Zwally and Fiegles (1994; dark blue lines). The multicolored right column is the corresponding percentage of melt days compared to total days from each of the four melt detection techniques and the AWS.**

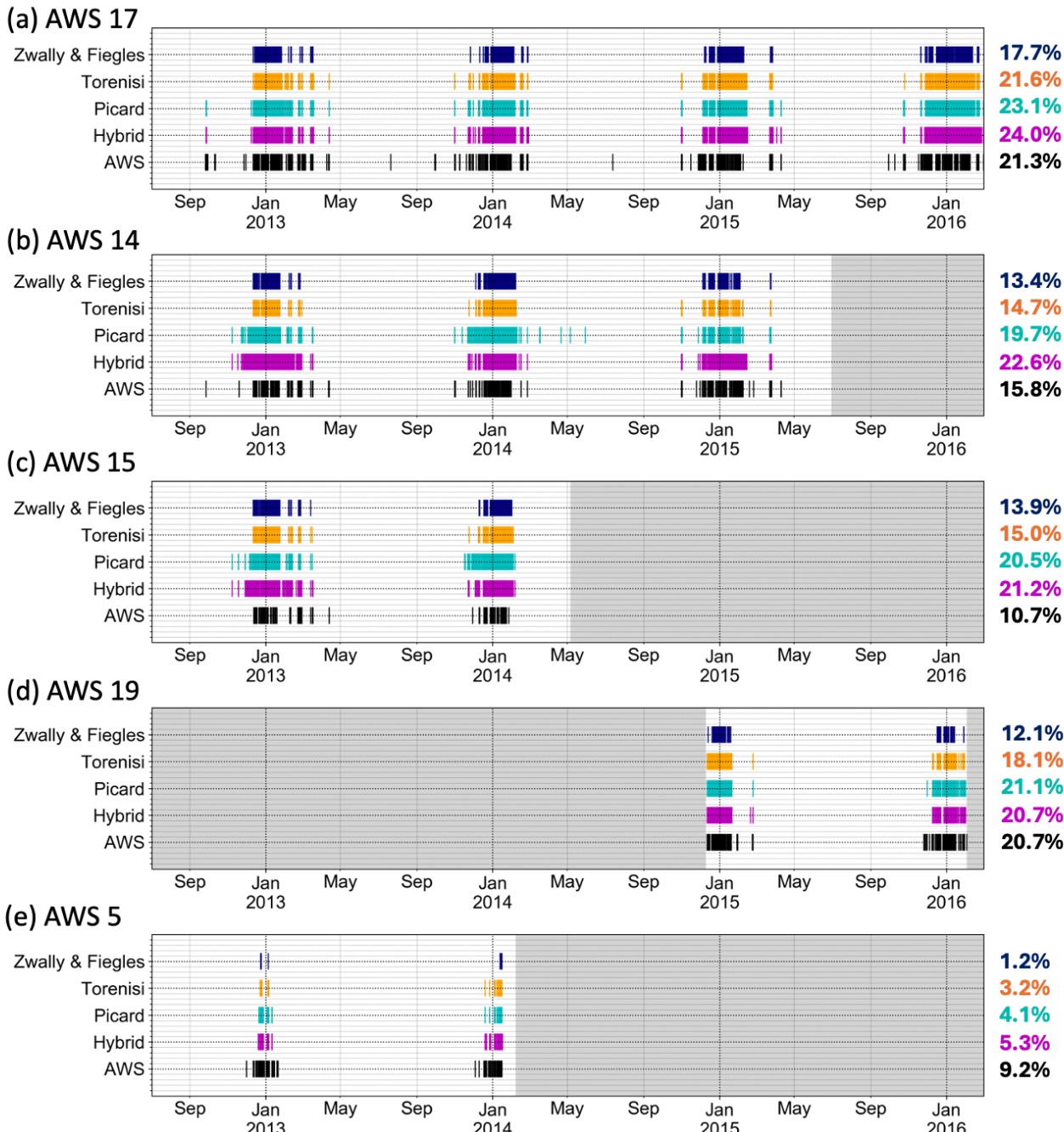

**Figure 6:** A comparison of melt detection between the Hybrid Method and three statistical thresholding techniques and AWS observations. Melt indicators from (a) AWS 17, (b) AWS 14, (c) AWS 15, (d) AWS 19, and (e) AWS 5. (a-e) Melt days when AWS surface energy balance melt > 0 (black lines). Melt days detected by Hybrid Method (purple


**lines). Melt days detected by Picard et al. (2022d; cyan lines), Torinesi et al., (2003; orange lines), and Zwally and Fiegles (1994; dark blue lines). The multicolored right column is the corresponding percentage of identified melt days compared to total days based on each of the four melt detection techniques and the AWS.**


| | # days | # AWS melt days | % Matching AWS | | | |
|---|---|---|---|---|---|---|
| | | | Hybrid Method | Picard et al. | Torenisi et al. | Zwally and Fiegles |
| **AWS 18** | 1498 | 407 | 89.19 | 89.25 | 87.38 | 86.38 |
| **AWS 17** | 1348 | 287 | 91.02 | 91.17 | 91.69 | 90.95 |
| **AWS 14** | 1094 | 173 | 90.86 | 91.77 | 93.78 | 93.42 |
| **AWS 15** | 674 | 72 | 88.87 | 88.72 | 93.32 | 94.07 |
| **AWS 19** | 421 | 87 | 94.30 | 94.77 | 95.01 | 91.45 |
| **AWS 5** | 586 | 54 | 95.73 | 94.88 | 94.03 | 91.98 |
| **AWS 11** | 2404 | 17 | 99.13 | 99.25 | 99.25 | 99.29 |
| **AWS 16** | 1265 | 10 | 99.21 | 99.21 | 99.21 | 99.21 |
| **Avg. (Weighted by # days)** | | | 94.21 | 94.34 | **94.64** | 94.10 |
| **Avg. (Weighted by # AWS melt days)** | | | 90.87 | 91.06 | **91.10** | 90.15 |


**Table 1: A table comparing the Hybrid Method, Picard et al. (2022d), Torinesi et al. (2003), and Zwally and Fiegles (1994) methods of melt day detection for eight AWS (AWS 18, 17, 14, 15, 19, 4, 11, and 16) to SEB-derived melt data. "Matching days" refers to the sum of both melt and non-melt days that match with AWS SEB-derived melt data.**

**"Mismatched non-melt days" refers to days when a technique identifies a non-melt day when that day is considered melt by AWS SEB-derived melt data. "Mismatched melt days" refers to days when a technique identifies a melt day when that day is considered dry by AWS SEB-derived melt data.**

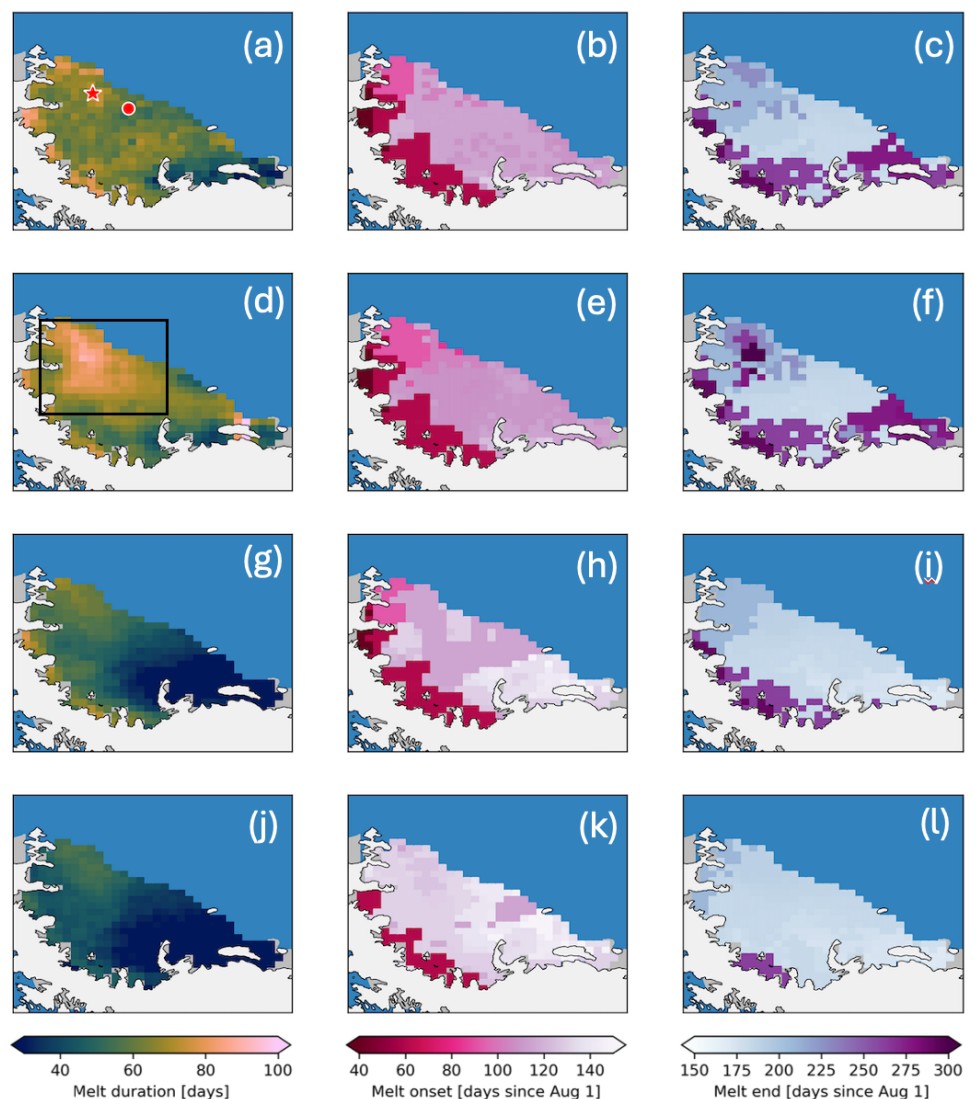

Figure 7: **The Hybrid Method melt detection and three statistical thresholding techniques across the Larsen C. Red star and red circle in (a) is the location of AWS 14 and AWS 15, respectively. Black box in (d) represents area of high melt duration detected by Picard et al. (2023d). (a, d, g, j) represents melt duration for the 2013-2014 melt season (number of melt days) using the Hybrid Method, Picard et al. (2022), Torensini et al. (2003), and Zwally and Fiegles (1994), respectively. (b, e, h, k) represents melt onset in days since August 1st 2013. (c, f, i, l) represents melt end in days since August 1st, 2013.**

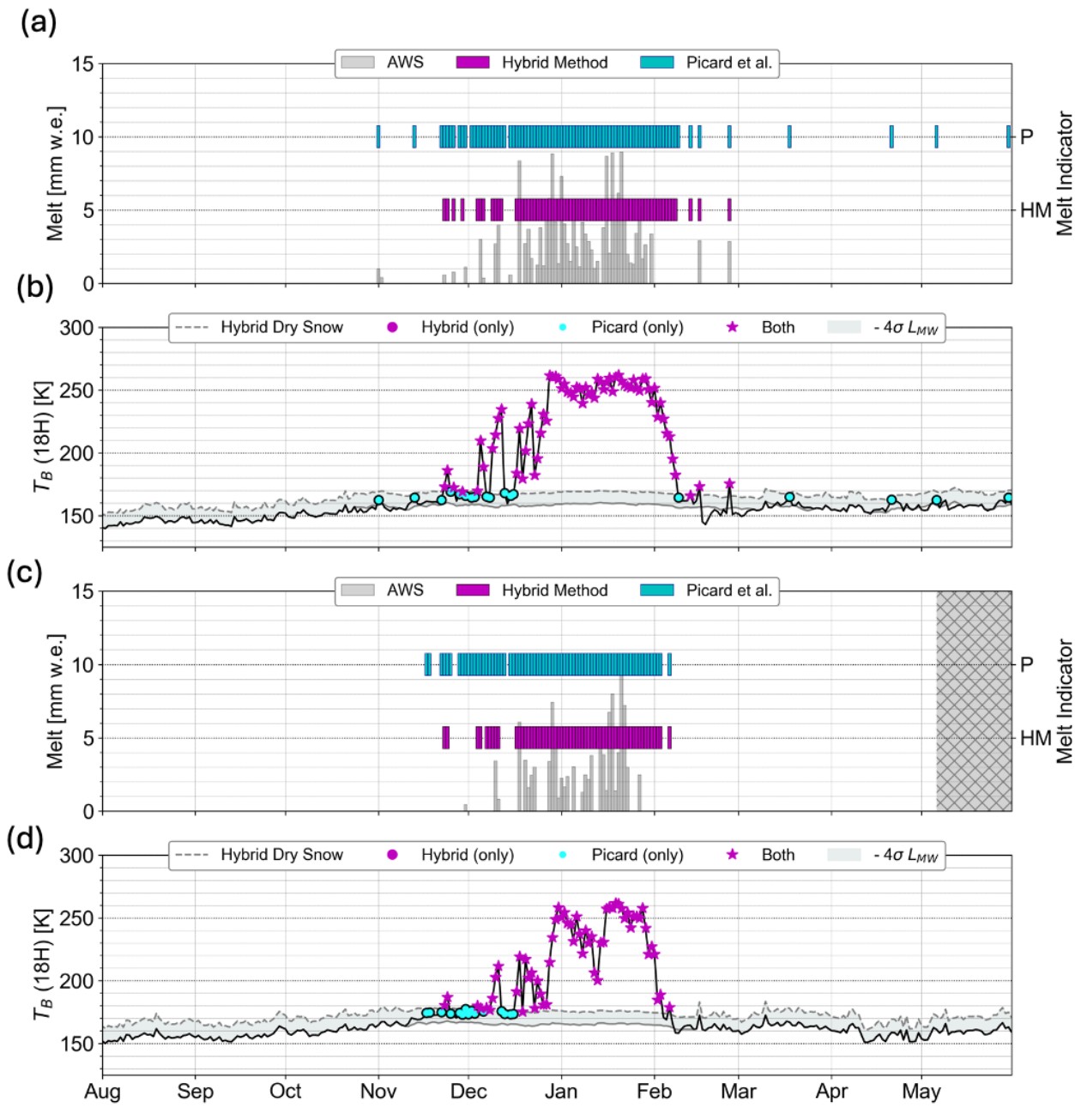

**Figure 8: Validation of melt detection using the Hybrid Method and Picard et al. (2022d) methods using observations at AWS 14 and AWS 15. (a) and (b) represent AWS 14 and (b) and (d) represent AWS 15. (a and c) AWS surface**

**energy balance melt volume (grey bars), melt days detected by Hybrid Method (purple lines), and melt days detected by Picard et al. (2022d; cyan lines). Hatched grey area represents period where AWS SEB data are not available. (c and d) Hybrid Method threshold (19H; dashed grey line) and AMSR-2 brightness temperature (19H; solid black line). Area between Hybrid Method threshold and Hybrid method dry snow (solid grey line) is shaded in light grey. Purple dots represent melt days detected by both the Hybrid Method and the Picard Method. Cyan dots represent melt days detected by only the Picard et al (2022d) method. Magenta stars represent melt days detected by both techniques. Magenta dots represent melt days detected by only the Hybrid Method.**

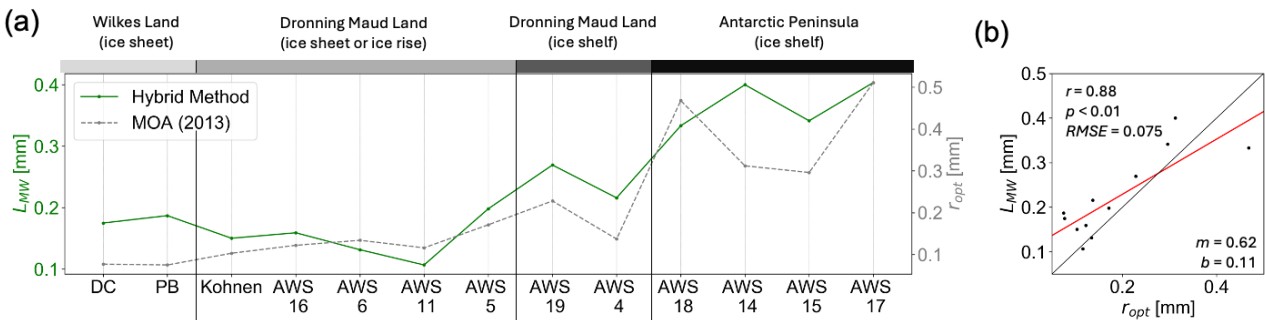

**Figure 9: Comparison between microwave grain size from the Hybrid Method and optical grain radius from MODIS 2013 Mosaic of Antarctica (MOA). (a) Microwave grain size ($L_{MW}$; green line) and optical grain radius from MOA ($r_{opt}$; dashed grey line) from 13 AWS (Haran et al., 2018). (b) Scatter plot of microwave grain size ($L_{MW}$) and optical grain radius ($r_{opt}$). Data from both the Hybrid Method and Mosaic of Antarctica are from 2013-10-27 to 2013-12-16.**

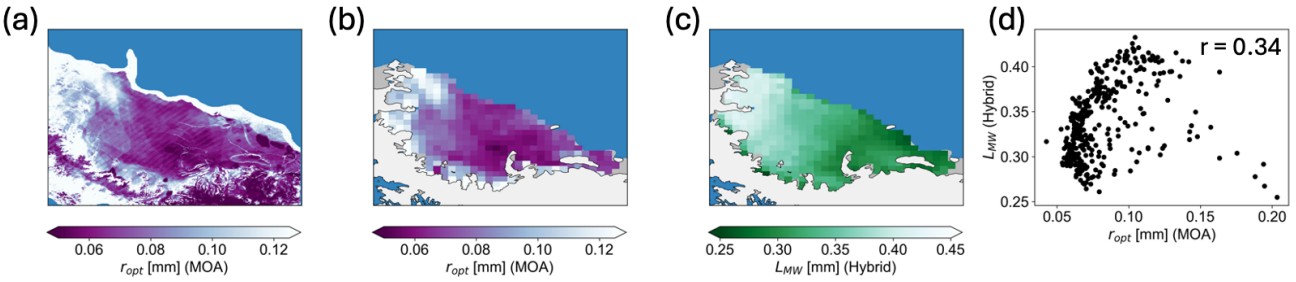

**Figure 10: Maps of the Larsen C to compare optical grain radius and microwave grain size from the Hybrid Method.** **(a) MODIS Mosaic of Antarctica (MOA; 2013) optical grain size ($r_{opt}$) (Haran et al., 2018). (b) MOA optical grain size ($r_{opt}$) re-gridded to 12.5 km x 12.5 km (Haran et al., 2018). (c) Microwave grain size ($L_{MW}$) from the Hybrid Method. (d) Scatter plot of microwave grain size ($L_{MW}$) and MOA optical grain radius ($r_{opt}$). Data from both the Hybrid Method and Mosaic of Antarctica are from 2013-10-27 to 2013-12-16.**

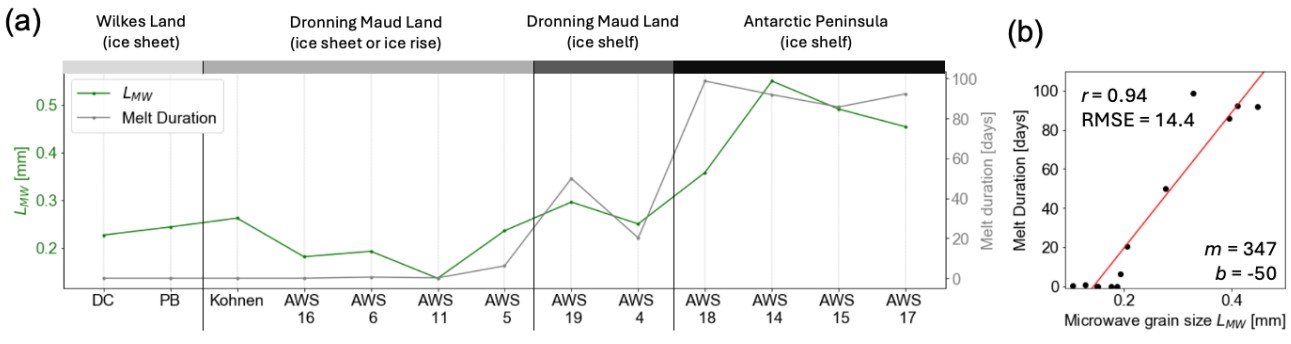

**Figure 11: Comparison between microwave grain size and melt duration both from the Hybrid Method. (a) Microwave grain size ($L_{MW}$; green line) and mean melt duration between mid-2012 and mid-2019 for 13 AWS. (b) Scatter plot of microwave grain size ($L_{MW}$) and mean melt duration. Red line represents linear regression.**

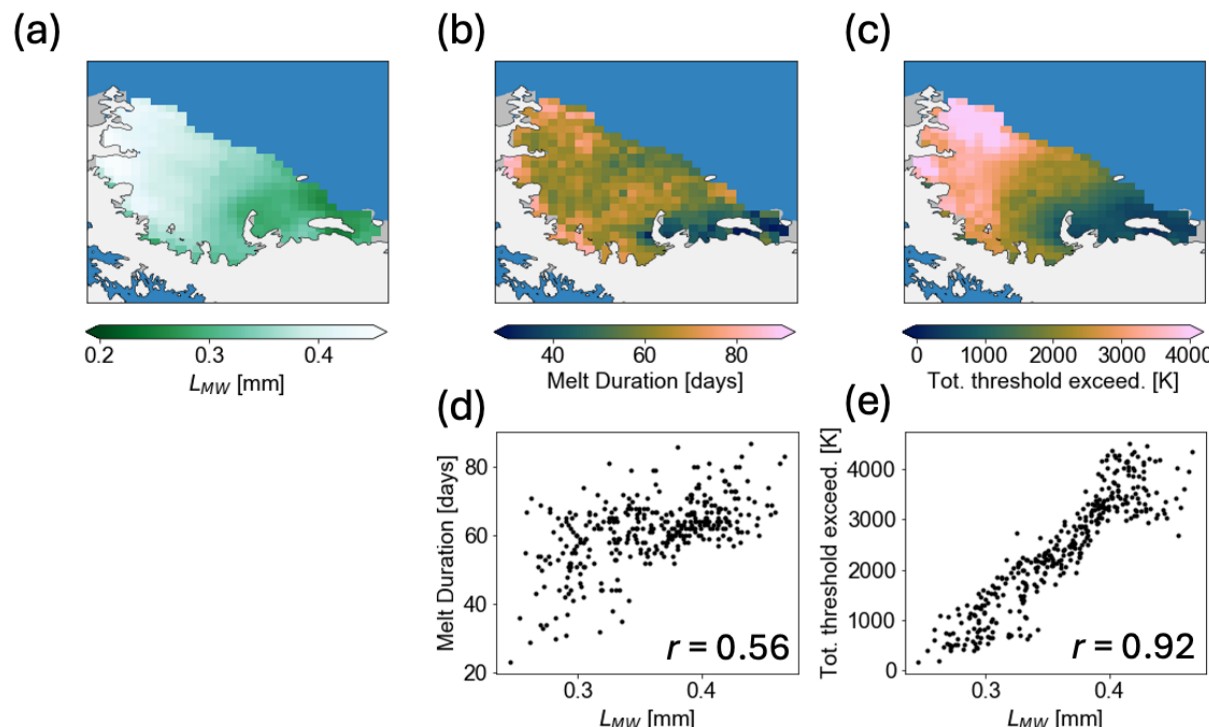

**Figure 12: Larsen C (a) Microwave grain size ($L_{MW}$) from the Hybrid Method (March 2014 to end of June 2014). (b) Melt duration for the 2013-2014 melt season from the Hybrid Method. (c) Total threshold exceedance, or the integral between the Hybrid Method's dynamic threshold and AMSR-2 observations for the 2013-2014 melt season. (d) Scatter plot of melt duration and microwave grain size ($L_{MW}$). (e) Scatter plot of total threshold exceedance and microwave grain size ($L_{MW}$).**