# Peer review of "A physics-based Antarctic melt detection technique: Combining AMSR-2, radiative transfer modeling, and firn modeling"

_The Cryosphere, 2023_

## Referee Comment (RC1)

**Review of Dattler et al. 'A physics-based Antarctic melt detection technique: Combining AMSR-2, radiative transfer modeling, and firn modeling'**
By Sophie de Roda Husman (S.deRodaHusman@tudelft.nl)

The manuscript introduces a novel algorithm, referred to as the Hybrid Method, designed for detecting surface melt in Antarctica. This approach leverages not only remote sensing data, as seen in conventional methods, but also integrates outputs from both the Community Firn Model and Snow Microwave Radiative Transfer Model. In this study, the threshold for discerning surface melt varies on a daily basis and is determined based on the variance of the correlation length over a specific time frame. Notably, this study also presents the correlation length as an output.

I consider this study to offer a highly valuable and innovative approach to melt detection. However, the manuscript could benefit from providing further clarity on the added value of the Hybrid Method. To strengthen its contribution, additional validation efforts should be undertaken. This might involve expanding the temporal scope for comparisons, offering less aggregated results, and broadening the spatial validation by incorporating more than two weather stations. These enhancements would contribute to a more robust evaluation of the method's effectiveness.

Strengths of this manuscript
Enhancing an algorithm with additional physical information is a noteworthy contribution to the existing literature. The authors have skillfully integrated numerous data products, and the clarity of the figures is notable. I found the flow chart in Figure 3 particularly effective in explaining the somewhat intricate methodology used in the manuscript.

Major comments
1. The **abstract of the manuscript requires refinement** as it lacks clarity regarding the research objectives, methodology, and key findings. It is crucial to distinctly highlight the primary advantage of the Hybrid Method over traditional thresholding techniques for melt detection. The assertion that "...this method is as accurate as previous statistically based thresholding techniques..." (L15) lacks persuasiveness and does not provide clear guidance on when and where to preferentially apply the Hybrid Method over established approaches.
2. This brings me to a critical point of consideration: upon reviewing the manuscript, I find myself uncertain about the **specific scenarios in which the Hybrid Method surpasses existing methodologies**. While Table 1 and Figure 8 offer valuable insights, they present aggregated data. It is crucial to discern when, within the melt season, and ideally, where in Antarctica, the Hybrid Method demonstrates superior performance, along with understanding the underlying reasons for this, as well as why alternative methods may fall short. Figure 7 could potentially provide assistance in this matter, although it hasn't been addressed in the manuscript as of now.
3. I find it crucial for the manuscript to **include an explanation of the physical interpretation of 'correlation length'**. Understanding the significance of this parameter and why it serves as an indicator of surface melt presence is

essential. A clear description of the correlation length would not only fortify the manuscript's overall strength but also enhance its readability.

4. Some of the important choices for developing the **melt detection threshold**, seemed arbitrary, examples are:
   a. $4\overline{\sigma_{p\_exp}}$ (L198, where is the 4 coming from?)
   b. +/- 7 days (L125, why 7 days?)
   c. 31-day window used to compute $4\overline{\sigma_{p\_exp}}$ (L197, why 31 days?)
   The manuscript would benefit from some further explanation of these numbers and potentially some sensitivity studies in which a variety of values is tested.

5. At present, the manuscript primarily relies on point-based comparisons for surface melt assessment involving AWS, the Hybrid Method, and other melt detection algorithms. Including **Antarctic-wide spatial maps** of the Hybrid Method, Picard's method (or other conventional melt detection algorithms), and the differences between the two regarding parameters like (1) number of melt days; (2) onset of the melt season; and (3) end of the melt season, would offer supplementary perspectives on the strengths and limitations of the Hybrid Method. Additionally, incorporating Antarctic-wide spatial maps of the newly introduced correlation length, along with snow grain size data from the Mosaic of Antarctica, would serve as valuable supplements to the manuscript.

6. The authors really challenged themselves by making a manuscript of which the **goal is two-fold**: creating a correlation length product and developing a new melt detection algorithm. I would suggest **adding some extra structure** to the manuscript (for example, adding clear sections to the result and discussion section in which both topics are presented).

7. The evaluation of the method is confined to just **two weather stations** (AWS17 and AWS18), which may be deemed **inadequate for comprehensive validation**. The manuscript mentions this selection was based on the significant overlap with AMSR-2 data, which left me somewhat puzzled. As far as I am aware, AMSR-2 data has been accessible since 2012. This prompts me to wonder why data from stations like AWS4, AWS5, AWS11, AWS14, AWS15, AWS19, and the Neumayer station are not utilized in Section 4.2.

8. In this study, both the **horizontal and vertical frequencies** of AMSR-2 are employed. I propose focusing solely on the horizontal frequency for the Hybrid Method, and the vertical frequency for the derivation of the correlation length. Now, sometimes one frequency (for example in Table 1), and sometimes both frequencies (for example in Figure 8) are used for the Hybrid Method. It remains unclear why both frequencies are employed for deriving surface melt presence and which one is the preferred choice for the Hybrid Method.

Minor comments

**Abstract**
- L10: I suggest to mention the underlying challenge here. Why does the scientific community necessitate a novel method for melt detection? It appears that you have well-founded reasons, as outlined in lines L33-L36.

Perhaps you could further emphasize that existing melt detection methods may falter in certain specific scenarios.

- L12: I suggest explicitly referring to your new method as the "Hybrid Method", and introducing the name of the method here. For example, by changing "… to create a hybrid method …" into "… we created a novel method, referred to as the Hybrid Method, …".
- L15: Quantifying the performance of the Hybrid Method compared to other methods (for example by using accuracy values of the Hybrid Method and other methods) would strengthen your point of introducing your novel method.
- L17: I would add the sentence about significant correlation (did you check for significance, by the way?) at the beginning of the abstract, thereby making the argument that correlation length is a good variable to use for thresholding surface melt presence.

**I. Introduction**

- L21: This is the first time the abbreviation 'AIS' is used, so please expand it to the *Antarctic Ice Sheet (AIS)* here (and not in L26).
- L34: Can you include a citation here? Also, consider referring to some studies that showed why and when certain conventional thresholding techniques fail. Maybe Johnson et al. (2020) could help here.
- L40: melt detection (instead of *melt-detection*)
- L44: I recommend providing a concise explanation of correlation length. Currently, it is only briefly mentioned in parentheses, yet it holds significant importance for the remainder of the manuscript.
- L45: Please introduce the abbreviations CFM and SMRT.
- L48: statistically-based (instead of *statistically based*)
- L49: What do you mean by 'intermediary calculations of snow microstructure'?

**II. Data & Models**

- L51: What is meant by 'AIS point'? Maybe replace it with something like 'Automatic weather stations'
- L53: I recently also used the data of Jakobs et al. (2020) and noted there was a small typo in Table 2 with the coordinates. Not sure if you used this table, but just to be sure, I believe the coordinates of AWS18 are (66.40; -63.37) instead of (66.40; -63.73).
- L57-L60: Why are there two starting and two ending days mentioned for AWS17 and AWS18? Can the last sentence (L59-L60) be removed?
- L66: Section 2.2 would benefit from some further elaborations. Some questions I still have:
  - Which period do you use?
  - What polarizations do you use (this becomes clear in Section 3, but might be nice to add here as well)?
  - From where do you download the data product mentioned in L66?
- L65: You refer to 18.7 GHz by shortening it to 19 GHz, while throughout the rest of the manuscript, 18 GHz is consistently used. In my suggestion, I recommend adhering to 19 GHz consistently, as it aligns with conventional terminology (as seen in works like Johnson et al., 2020).

- L72: add the source of the data product Mosaic of Antarctica
- L72: What does it mean for your final melt product that CFM underestimates snow grain size (compared to Mosaic of Antarctica)?
- L82: Mention what the exact output of CFM is (snow density and temperature, right?).
- L84: Consider replacing thermal emission with *brightness temperature* and backscattering with *backscatter intensity,* for consistency.

**III. Methodology**
- Consider adding a section (3.1) on 'deriving surface melt presence with conventional thresholding techniques'. The melt presence derived from Picard et al., Torinesi et al., and Zwally and Fiegles are presented in the figures, but a short explanation of the methods is missing.
- L102-103: You mention that the CFM grain size profiles can be converted into correlation length with 'some uncertainty'. Can you add some additional explanation here?
- L112: What 'outside source' is meant here?
- L134: Replace 'these two polarizations' with 'horizontal and vertical polarizations'
- L139: How do you select a 'high' and 'low' correlation length value? And what is the physical meaning of a 'high' and 'low' correlation length? I can imagine that for a low correlation length, the snow properties are more heterogeneous. Therefore, I expect conditions to be more conducive to melt, as local heterogeneities and variations in temperature or impurity concentration can lead to localized melting processes.
- L144: It's a bit confusing that terms like 'many times' and 'narrower bands' aren't quantified here. While this information is provided later in the manuscript (L157-159), I would suggest revising this section to ensure a smoother flow of the narrative, without the need to constantly refer back and forth.
- L148: You might want to consider giving more intuitive names to Process 1, Process 2, and Process 3, in the text. For example, you could label them as follows: "Step 1: Calculate Correlation Length", "Step 2: Assign Melt Days / Dry Days", and "Step 3: ??", as you do in Figure 3. By the way, it appears that "Step 3" (or Process 3) is not represented in Figure 3.
- L190: What type of 'other information' is meant here?
- L191: To maintain consistency, I recommend using 'melt' and 'non-melt' days, as you have done in this line, rather than 'melt' and 'dry' days, which are used elsewhere in section 3.1 and Figure 3.
- L198: You could add 'hereafter referred to as $4\overline{\sigma_{p\_exp}}$ *or dynamic threshold*).

**IV. Results**
- L215-L222: These lines introduce a significant amount of supplementary information, which may divert attention from the main narrative. Could this be included in the method section rather than the results?

- L224: Wouldn't it be more appropriate to introduce SSA in the data section? And also the next line (L225), seems to be more fitting for the method section that for the results.
- L245: Consider referring to AWS melt, instead of SEB melt, in the section title.
- L258: What is meant by 'increases across each of the four austral summers shown'? I was expecting the average correlation length per melt season to progressively rise compared to the preceding season, but this trend isn't clearly evident in Figure 5b.
- L265: Also Torinesi's method yields an accuracy of 91.7%, correct? While you refer to it as the 'second best technique', from my perspective, it actually attains the lowest accuracy (tied with Zwally & Fiegles' method). Could you clarify if I'm overlooking something in this regard?
- L290: As I highlighted in the 'Major comments' section, I believe the manuscript would greatly benefit from including a comparison with additional weather stations. Comparing the Hybrid Method with other statically-based techniques may not provide substantially new insights. Given that statistically-based methods are acknowledged to have limitations in accurately detecting melt (which prompted the development of this new method, as you state in the introduction section), what meaningful insights do we gain from this comparison?
- L295: I would consistently use capitals for your new method, using 'Hybrid Method' rather than 'hybrid method'. Currently, both are employed (sometimes also in one sentence, for example in L599).
- L301: Could you explain why this is not the case for AWS15?
- L309: I would suggest to introduce MOA in the method section.
- L303-L322: This appears somewhat abrupt and disrupts the flow of the narrative. Perhaps consider creating a new subsection in the results dedicated to the evaluation of the generated correlation length product?
- L314: I'd be interested in seeing a spatial map that compares MOA's optical grain size with your correlation length product. Additionally, a scatter plot illustrating the correlation between both products would be informative.

**Discussion**
- General comment: I recommend using distinct titles for the topics discussed in the subsection to enhance readability.
- L325: Consider incorporating more specific statistical terminology, instead of mentioning that the methods are 'tied'? This might enhance the clarity of the comparison.
- L338: Which two sites do you mean?
- L339: Can you quantify this statement?
- L363: What do you mean by 'resulting overestimate of emissivity'?
- L379: What do you mean by 'that site'? Are you referring to a specific location or is this a general statement?
- L379: Did you test that there was a significant correlation? This might be good to add to the methods section.
- L399: In my (somewhat limited – sorry about that!) understanding of correlation length, I thought that surface melt could also result in a decrease

in correlation length. As the ice undergoes a phase change from solid to liquid, this transition can introduce irregularities in the ice structure, such as the formation of pores, cracks, or fractures filled with meltwater. I thought this process could reduce the correlation length. I assume this is incorrect, right?

**Conclusion**
- Looks good!

**Author contributions**
- Shouldn't it be MS instead of CS?

**Data availability**
- Really nice that the data are shared on Zenodo.
- Could you provide information about the sources from which you obtained all the data products utilized in this study?

**Figures**
- **Figure 1:** Consider adding a title to the legend, indicating something like 'AWS located on:' for clarity. Currently, it might be a bit confusing at first glance.
- **Figure 1:** In the caption, I think 'AIS' should be replaced by 'AWS'. Or, even better, avoid abbreviations in the figure captions, and replace by 'Antarctic Weather Stations'.
- **Figure 1:** The shades of blue and cyan appear quite similar to me. Please consider replacing one of the colors?

- **Figure 2:** Nice figure! One very small comment, maybe you could add one line with a general description at the beginning of the caption?

- **Figure 3:** Nice figure!
- **Figure 3:** Process 3 is mentioned in the text (L148), why is this step not presented in the figure?
- **Figure 3:** What are $TB_1$ and $TB_2$? Do they refer to horizontal and vertical polarizations, or the 'low' and 'high' brightness temperatures coming from the 'low' and 'high' correlation lengths?

- **Figure 4:** Again, I struggle to see the difference between blue and cyan.
- **Figure 4:** Could you clarify what is referred to as "NIR SSA" in the caption? Is this data product discussed in the text? Additionally, consider using more intuitive labels, such as indicating the 4 and 10 meter penetration depths, instead of " REFL. SP1" and "REFL. SP2".
- **Figure 4:** In subpanels (c) and (d), I cannot see the AMSR-2 line. Consider increasing the linewidth, so you see it is overlapping with the other lines.
- **Figure 4:** I would suggest not abbreviating 'BC' in the caption, as there are already numerous abbreviations in use, which can hinder readability.
- **Figure 4:** In panel (c) and (d), the legend states that $-4\overline{\sigma_{p\_exp}}$ is shown. Shouldn't this be $+4\overline{\sigma_{p\_exp}}$? And in panel (b), both + and − are shown. I got a

bit confused here, when to use – (like in panels c and d), and when to use – and $+ 4\overline{\sigma_{p\_exp}}$ (like in panel b)?

- **Figure 4:** Please consider merging panels (c) and (d). If they are plotted separately, the distinction isn't clear to me. Alternatively, displaying only panel (d) and (e) could effectively convey the message.
- **Figure 4:** In panel (a), 18H is depicted, while in panels (c) and (d), 18V is utilized. It is not entirely clear to me why 18H is employed for melt detection, and 18V for establishing the threshold.
- **Figure 4:** In panel (b), I find it challenging to discern the median filter. Additionally, could you clarify the purpose of using the median filter?

- **Figure 5** and **Figure 6:** I'm having difficulty comparing the melt days obtained from the Hybrid Method (panel d) with those derived from the statistically-based method (panel a). Could you consider plotting them in a single figure for easier comparison?

- **Figure 7:** This figure is not mentioned in the text, please include it in the results section.

- **Figure 9:** I recommend placing this as the initial figure, possibly in the form of a scatter plot where you plot % melt days against correlation length. This visualization would effectively demonstrate that correlation length is a reliable indicator for surface melt, thereby bolstering your argument for using this method.

**Used references**

- Johnson, A., Fahnestock, M., & Hock, R. (2020). Evaluation of passive microwave melt detection methods on Antarctic Peninsula ice shelves using time series of Sentinel-1 SAR. Remote Sensing of Environment, 250, 112044.

---

## Author Comment (AC2)

**A physics-based Antarctic melt detection technique: Combining AMSR-2, radiative transfer modeling, and firn modeling**
Marissa Dattler, Brooke Medley, C. Max Stevens

**Review #1**
Review of Dattler et al. 'A physics-based Antarctic melt detection technique: Combining AMSR-2, radiative transfer modeling, and firn modeling'

The manuscript introduces a novel algorithm, referred to as the Hybrid Method, designed for detecting surface melt in Antarctica. This approach leverages not only remote sensing data, as seen in conventional methods, but also integrates outputs from both the Community Firn Model and Snow Microwave Radiative Transfer Model. In this study, the threshold for discerning surface melt varies on a daily basis and is determined based on the variance of the correlation length over a specific time frame. Notably, this study also presents the correlation length as an output.

I consider this study to offer a highly valuable and innovative approach to melt detection. However, the manuscript could benefit from providing further clarity on the added value of the Hybrid Method. To strengthen its contribution, additional validation efforts should be undertaken. This might involve expanding the temporal scope for comparisons, offering less aggregated results, and broadening the spatial validation by incorporating more than two weather stations. These enhancements would contribute to a more robust evaluation of the method's effectiveness.

> Thank you for your review and consideration of our manuscript; it is very much appreciated. Based on your comments, we are making five main changes to the manuscript:
>
> (1) Broadening spatial coverage by running SMRT across the Larsen C ice shelf for the 2013-2014 melt season, comparing established statistical thresholding techniques and validating correlation length.
> (2) Broadening the spatial validation by incorporating eight AWS instead of two.
> (3) Offering less aggregated results by introducing a new figure in which we compare in detail each of the now eight AWS to SEB observations and established statistical thresholding techniques.
> (4) Expanding the temporal scope for comparisons by extending the time frame of our AWS melt analysis to include six melt seasons (July 2012-May 2019).
> (5) Reframing the hybrid method as not a replacement for quicker statistically based techniques, but as a stepping-stone towards melt quantification and quantification of other relevant snow parameters on the Antarctic Ice Sheet.

Strengths of this manuscript
Enhancing an algorithm with additional physical information is a noteworthy contribution to the existing literature. The authors have skillfully integrated numerous data products, and the clarity of the figures is notable. I found the flow chart in Figure 3 particularly effective in explaining the somewhat intricate methodology used in the manuscript.

Thank you for your comments.

Major comments
1. The abstract of the manuscript requires refinement as it lacks clarity regarding the research objectives, methodology, and key findings. It is crucial to distinctly highlight the primary advantage of the Hybrid Method over traditional thresholding techniques for melt detection. The assertion that "...this method is as accurate as previous statistically based thresholding techniques..." (L15) lacks persuasiveness and does not provide clear guidance on when and where to preferentially apply the Hybrid Method over established approaches.

We updated the abstract to better address these concerns, clarifying now the hybrid method works as well as providing guidance on the use of the hybrid method:

**Abstract.** Surface melt on ice shelves has been linked to hydrofracture and subsequent ice shelf breakup. Since the 1990s, scientists have been using microwave radiometers to detect melt on ice shelves and ice sheets by applying various statistical thresholds to identify significant increases in brightness temperature that are associated with melt. In this study, instead of using a fixed threshold, we force the Snow Radiative Transfer Model with outputs from the Community Firn Model to create a dynamic, physics-based threshold for melt. In the process, we also hybridize our method to statistical techniques and produce microwave grain size information in the process. We run this "hybrid method" on 13 sites over the Antarctic Ice Sheet as well as across the Larsen C ice shelf. Compared to surface energy balance measurements from automatic weather stations, this method is as accurate as previous statistically based thresholding techniques. However, this method is far more computationally expensive than statistical techniques and they both can detect the dramatic increases in brightness temperature associated with melt. Rather than replacing statistical thresholds with the hybrid method, we recommend using the hybrid method in studies where melt volume or grain size is of interest. Specifically, we show that our microwave grain size output correlates with melt frequency. We show in this study that the hybrid method can be used to (a) model brightness temperatures of Antarctic snow and (b) derive a measure of grain size; therefore, it is an important step forwards towards melt quantification rather than melt detection.

We also include more relevant information in the discussion (L403):

> In this study, we showed that there were relatively minor differences in melt detection between the hybrid method and the other statistical thresholding techniques. This finding is somewhat unsurprising, as brightness temperature increases dramatically between the presence of liquid water on an ice sheet versus dry snow conditions. Both detailed, physics-based modeling such as the hybrid method as well as statistical thresholding techniques can exploit this relationship to identify melt days versus dry days. An important aspect of the hybrid method that is worth noting is that is highly computationally expensive. Running SMRT multiple times to complete our inversion for each day and each AMSR-2 grid cell requires a significant amount of computation time. Therefore, expanding this technique to be run ice sheet-wide is a computational challenge. For these reasons, statistically based techniques are still an important tool for melt detection.

2. This brings me to a critical point of consideration: upon reviewing the manuscript, I find myself uncertain about the specific scenarios in which the Hybrid Method surpasses existing methodologies. While Table 1 and Figure 8 offer valuable insights, they present aggregated data. It is crucial to discern when, within the melt season, and ideally, where in Antarctica, the Hybrid Method demonstrates superior performance, along with understanding the underlying reasons for this, as well as why alternative methods may fall short. Figure 7 could potentially provide assistance in this matter, although it hasn't been addressed in the manuscript as of now.

This is a great point that we are clarifying in the updated manuscript:

1) We are adding an additional figure, similar Figure 7b/d. This figure will include 6 more AWS. It will show that to show that the hybrid method, like the other statistically based methods, appears to work best for days and locations where there is significant melt, such as the Larsen C. It captures more melt in the beginning/end of the melt season.
2) We will add information to the discussion section clarifying that there is a much bigger difference between how an AWS measures melt and any one of the hybrid method or existing methodologies. We will clarify that the intent of the hybrid method is as a steppingstone towards melt volume.

3. I find it crucial for the manuscript to include an explanation of the physical interpretation of 'correlation length'. Understanding the significance of this parameter and why it serves as an indicator of surface melt presence is essential. A clear description of the correlation length would not only fortify the manuscript's overall strength but also enhance its readability.

We are adding a paragraph to section 2.3 on "correlation length", or what we now refer to as "microwave grain size" after a suggestion from Reviewer #2. Microwave grain size is a unifying term as it can be computed from other microwave techniques. For additional clarification, we are adding more description section on microwave grain size as shown below (added to L89).

Furthermore, snow microstructure model we chose to implement is the exponential snow microstructure model while using IBA within SMRT (SMRT-IBA) because the exponential model considers grain size a single parameter (Picard et al., 2018). Given that we are using an exponential snow microstructure model, this may be referred to as the exponential correlation length ($L_C$). In this paper, we refer to this value as "microwave grain size" ($L_{MW}$), a unifying concept that allows for comparison to other to snow microstructure parameters using other models and techniques (Picard et al., 2022b). Microwave grain size is a length scale for snow microstructure (Picard et al., 2022b). Microwave grain size ($L_{MW}$) is proportional to the Porod length ($L_P$), which relates snow density ($\rho$) and optical grain diameter ($d_{opt}$) in Mätzler (2002; Picard et al., 2022b):

$$L_{MW} = L_C = \alpha L_P = \alpha \, (2/3) \, (1 - \rho/\rho_{ice}) \, d_{opt} \hspace{2cm} (1)$$

Equation (1) shows that (a) microwave grain size is proportional to grain diameter, (b) larger optical grain diameter are associated with higher microwave grain sizes, and (c) higher densities are associated with lower microwave grain sizes. Therefore, microwave grain size is a complicated snow microstructural input into SMRT-IBA that relates to both grain diameter and density.

4. Some of the important choices for developing the **melt detection threshold**, seemed arbitrary, examples are:
   a. $4\overline{\sigma_{p\_exp}}$ (L198, where is the 4 coming from?)
   b. +/- 7 days (L125, why 7 days?)
   c. 31-day window used to compute $4\overline{\sigma_{p\_exp}}$ (L197, why 31 days?)

   The manuscript would benefit from some further explanation of these numbers and potentially some sensitivity studies in which a variety of values is tested.

Thank you for your recommendation. We will go into more detail in the manuscript as it applies to the three choices you brought up:

(a & c) We will include a figure relating $4\overline{\sigma_{p_{exp}}}$ (now $4\overline{\sigma_{L_{MW}}}$) to the uncertainty that results from potential errors within CFM. We chose to use $4\overline{\sigma_{p_{exp}}}$ at 31-day intervals as it is a close approximation to the uncertainty in our microwave grain sizes/correlation lengths. This is an ideal option over requiring the user to recompute model uncertainty on every iteration as that would double the runtime of this algorithm. We are including this explanation in Methodology section 3.2.

In a world where our CFM and SMRT modeling were perfect, our dry snow brightness temperatures alone could act as a threshold for melt without adding a threshold. However, there is still uncertainty in our methodology, thus the addition to the threshold cannot be zero. On the other hand, this perturbation in microwave grain size/correlation length results a thresholding difference of 5-10 K. This value is substantially less than 20 K, which is the value used in Picard et al., 2020.

(b) Examining all melt in combined AWS 4, 5, 14, 15, 17, 18, and 19, over 80% of observed melt days fall within +/- 7 days of Picard et al. (2020) melt days. We now identify in the Methodology section 3.1 that this is the reason why we chose +/- 7 days.

5. At present, the manuscript primarily relies on point-based comparisons for surface melt assessment involving AWS, the Hybrid Method, and other melt detection algorithms. Including Antarctic-wide spatial maps of the Hybrid Method, Picard's method (or other conventional melt detection algorithms), and the differences between the two regarding parameters like (1) number of melt days; (2) onset of the melt season; and (3) end of the melt season, would offer supplementary perspectives on the strengths and limitations of the Hybrid Method. Additionally, incorporating Antarctic-wide spatial maps of the newly introduced correlation length, along with snow grain size data from the Mosaic of Antarctica, would serve as valuable supplements to the manuscript.

Unfortunately, we are not able at this time to run the hybrid method across the Antarctic Ice Sheet because it is extremely computationally expensive to do so. Instead, to address this concern the effects of spatial patterns in melt, we are **running the hybrid method across the Larsen C ice shelf** from mid 2013 to mid 2014. We will show (1) the number of melt days calculated by the hybrid method versus Picard method, (2) onset of melt season (3) end of melt season. This will show that the hybrid method present similar results to the Picard et al. method, demonstrating that radiative transfer models can successfully detect melt. This allows us to how well the Hybrid method works across a strong spatial gradient of melt. We will present a map of microwave grain size across the Larsen C, showing how it correlates to melt, and comparing it to the Mosaic of Antarctica.

The reason why we are currently unable to run the full Antarctic Ice Sheet is because inverting SMRT and running it over many layers takes a huge amount of computation time. To make this issue clear to readers, we added a section in the discussion to address this limitation on L403:

In this study, we showed that there were relatively minor differences in melt detection between the hybrid method and the other statistical thresholding techniques. This finding is somewhat unsurprising, as brightness temperature increases dramatically between the presence of liquid water on an ice sheet versus dry snow conditions. Both detailed, physics-based modeling such as the hybrid method as well as statistical thresholding techniques can exploit this relationship to identify melt days versus dry days. An important aspect of the hybrid method that is worth noting is that is highly computationally expensive. Running SMRT multiple times to complete our inversion for each day and each AMSR-2 grid cell requires a significant amount of computation time. Therefore, expanding this technique to be run ice sheet-wide is a computational challenge. A small change to improve speed would be implanting the Brent method instead of the Secant method during our inversion to calculate microwave grain size (Brent, 1973). For these reasons, statistically based techniques are still an important tool for melt detection.

6. The authors really challenged themselves by making a manuscript of which the goal is two-fold: creating a correlation length product and developing a new melt detection algorithm. I would suggest adding some extra structure to the manuscript (for example, adding clear sections to the result and discussion section in which both topics are presented).

We are rearranging our manuscript such that the results and discussion are broken up into sections about melt detection and sections about microwave grain size, as follows:

4 Results
4.1 Validation of dry snow zone
4.2 Melt detection
   4.2.1 AWS sites: Comparison to SEB-derived melt data and established methods
   4.2.2 Larsen C ice shelf: Comparison to established methods
4.3 Microwave grain size
  4.3.1 AWS sites: Microwave grain size, melt frequency, and Mosaic of Antarctica
  4.3.2 Larsen C ice shelf: Microwave grain size, melt frequency, and Mosaic of Antarctica

5 Discussion
5.1 Melt detection
5.2 Microwave grain size

7. The evaluation of the method is confined to just two weather stations (AWS17 and AWS18), which may be deemed inadequate for comprehensive validation. The manuscript mentions this selection was based on the significant overlap with AMSR-2 data, which left me somewhat puzzled. As far as I am aware, AMSR-2 data has been accessible since 2012. This prompts me to wonder why data from stations like AWS4, AWS5, AWS11, AWS14, AWS15, AWS19, and the Neumayer station are not utilized in Section 4.2.

We extended our time series so that we could add AWS4, AWS5, AWS11, AWS14, AWS15, and AWS19* in an additional figure to this section. This is explained as an answer to comment #2. It will allow us to compare melt derived from SEB observations to the hybrid method run at these locations. It will also allow us to intercompare all three melt detection algorithms, in order to show the timing in the melt season where the hybrid method differs from the statistical techniques.

*Unfortunately, we had difficulty locating the SEB melt data from the Neumayer station when we run these models. Since we are already presenting 8 AWS stations, we decided to exclude this site as it would not add much additional information towards the goals of this paper.

8. In this study, both the horizontal and vertical frequencies of AMSR-2 are employed. I propose focusing solely on the horizontal frequency for the Hybrid Method, and the vertical frequency for the derivation of the correlation length. Now, sometimes one frequency (for example in Table 1), and sometimes both frequencies (for example in Figure 8) are used for the Hybrid Method. It remains unclear why both frequencies are employed for deriving surface melt presence and which one is the preferred choice for the Hybrid Method.

We agree with your proposal, and we are now only showing results for the horizontal polarization for melt, and only showing results for the vertical polarization for correlation length. Melt frequency from 18V and microwave grain size/correlation length will be removed from Figure 8 and Figure 9.

Minor comments
Abstract
• L10: I suggest to mention the underlying challenge here. Why does the scientific community necessitate a novel method for melt detection? It appears that you have well-founded reasons, as outlined in lines L33-L36. Perhaps you could further emphasize that existing melt detection methods may falter in certain specific scenarios.

Based on earlier suggestions, we changed the abstract to include:

frequency. We show in this study that the hybrid method can be used to (a) model brightness temperatures of Antarctic snow and (b) derive a measure of grain size; therefore, it is an important step forwards towards melt quantification rather than melt detection.

• L12: I suggest explicitly referring to your new method as the "Hybrid Method", and introducing the name of the method here. For example, by changing "... to create a hybrid method ..." into "... we created a novel method, referred to as the Hybrid Method, ...".

Adjusted to "...to create a novel, physics-based threshold for melt called the 'Hybrid Method'"

• L15: Quantifying the performance of the Hybrid Method compared to other methods (for example by using accuracy values of the Hybrid Method and other methods) would strengthen your point of introducing your novel method.

> We adjusted the abstract to explain that the Hybrid Method isn't necessarily better at melt detection than any one statistically-based technique; its utility comes from its physics-based nature and flexibility to be modified for melt quantification techniques.

• L17: I would add the sentence about significant correlation (did you check for significance, by the way?) at the beginning of the abstract, thereby making the argument that correlation length is a good variable to use for thresholding surface melt presence.

> Included that this correlation is significant (shown on L320 that $p < 0.01$).

I. Introduction

• L21: This is the first time the abbreviation 'AIS' is used, so please expand it to the Antarctic Ice Sheet (AIS) here (and not in L26).

> Defined on L21.

• L34: Can you include a citation here? Also, consider referring to some studies that showed why and when certain conventional thresholding techniques fail. Maybe Johnson et al. (2020) could help here.

> Thank you for the suggestion; we are including a reference to Johnson et al. (2020) here (L34).

• L40: melt detection (instead of melt-detection)

> Switched to melt detection.

• L44: I recommend providing a concise explanation of correlation length. Currently, it is only briefly mentioned in parentheses, yet it holds significant importance for the remainder of the manuscript.

> We include a new section on correlation length/microwave grain size, shown as a response to your major comment #3.

• L45: Please introduce the abbreviations CFM and SMRT.

> We include (CFM) and (SMRT).

• L48: statistically-based (instead of statistically based)

> Changed all "statistically based" to "statistically-based" throughout the paper.

• L49: What do you mean by 'intermediary calculations of snow microstructure'?

> Specified as follows:
>
> data as well as results from statistically based techniques. At thirteen Antarctic sites and the Larsen C, we analyze the frequency at which we detect melt days, as well as our intermediary calculations of snow grain size during our melt detection technique.

II. Data & Models

• L51: What is meant by 'AIS point'? Maybe replace it with something like 'Automatic weather stations'

> Changed to AWS.

• L53: I recently also used the data of Jakobs et al. (2020) and noted there was a small typo in Table 2 with the coordinates. Not sure if you used this table, but just to be sure, I believe the coordinates of AWS18 are (66.40; -63.37) instead of (66.40; -63.73).

      According to Table 2 in Jakobs et al. (2020) a longitude of -63.73 is correct.
      (doi:10.1017/jog.2020.6)

• L57-L60: Why are there two starting and two ending days mentioned for AWS17 and AWS18? Can the last sentence (L59-L60) be removed?

      All dates set to 2012-07-02 to 2019-05-31, as we extended the study for all sites to
      include these dates.

• L66: Section 2.2 would benefit from some further elaborations. Some questions I still have:

      o Which period do you use?

            Including that we use 2012-07-02 to 2019-05-31.

      o What polarizations do you use (this becomes clear in Section 3, but might be nice to add here as well)?

            Including that we use 19 GHz for melt detection and 19 GHz for microwave
            grain size.

      o From where do you download the data product mentioned in L66?

            Including reference to: doi:10.5067/RA1MIJOYPK3P

• L65: You refer to 18.7 GHz by shortening it to 19 GHz, while throughout the rest of the manuscript, 18 GHz is consistently used. In my suggestion, I recommend adhering to 19 GHz consistently, as it aligns with conventional terminology (as seen in works like Johnson et al., 2020).

      Changed to 19 GHz/19V/19H throughout the manuscript.

• L72: add the source of the data product Mosaic of Antarctica

      Added citation for doi:10.5067/RNF17BP824UM

• L72: What does it mean for your final melt product that CFM underestimates snow grain size (compared to Mosaic of Antarctica)?

      We now specify on this line that we do not use any snow grain size information from
      the CFM; therefore, it does not impact our results.

• L82: Mention what the exact output of CFM is (snow density and temperature, right?).

      We added "The output from CFM that we use is snow temperature and density with
      depth."

• L84: Consider replacing thermal emission with brightness temperature and backscattering with backscatter intensity, for consistency.

      Switched to "brightness temperature and backscatter intensity".

III. Methodology

• Consider adding a section (3.1) on 'deriving surface melt presence with conventional thresholding techniques'. The melt presence derived from Picard et al., Torinesi et al., and Zwally and Fiegles are presented in the figures, but a short explanation of the methods is missing.

      Added section 3.1:

**3.1 Deriving surface melt presence with conventional thresholding techniques**

In our study, we hybridize our method to a statistically-based method presented in Picard et al. (2022). In this method, the mean of June to September for a grid cell acts as a "dry brightness temperature" with melt being considered for any brightness temperature that exceeds the dry brightness temperature plus 20 K. 20 K above dry brightness temperature a relatively low threshold, meaning that this technique produces many melt days. We also compare our hybrid method to the method introduced by Torinesi et al. (2003). Torinesi et al. (2003) introduced a recursive method that involves taking the annual mean brightness temperature (April 1st to March 31st) and removing more and more melt days by taking the mean and standard deviation of the remaining melt days with each iteration. Finally, we compare to a method presented in Zwally and Fiegles (1994), where the threshold for melt is taken to be 30 K above mean brightness temperature.

• L102-103: You mention that the CFM grain size profiles can be converted into correlation length with 'some uncertainty'. Can you add some additional explanation here?

> size profiles, which can be converted to microwave grain size following Mätzler (2002), which is approximated for the Alps. As closer approximation can be made using Picard et al. (2022). However, these profiles from CFM are highly dependent on

• L112: What 'outside source' is meant here?

> Changed to a source "external to the hybrid method".

• L134: Replace 'these two polarizations' with 'horizontal and vertical polarizations'

> Replaced.

• L139: How do you select a 'high' and 'low' correlation length value? And what is the physical meaning of a 'high' and 'low' correlation length? I can imagine that for a low correlation length, the snow properties are more heterogeneous. Therefore, I expect conditions to be more conducive to melt, as local heterogeneities and variations in temperature or impurity concentration can lead to localized melting processes.

> We changed the name from correlation length to microwave grain size to make it clear that this is a high and low estimate of the grain size. We're including an explanation in Data & Models section about grain size which hopefully will make it more clear that we're talking about the correlation length of the snow microstructure, not correlation lengths of macroscopic snow properties.

• L144: It's a bit confusing that terms like 'many times' and 'narrower bands' aren't quantified here. While this information is provided later in the manuscript (L157-159), I would suggest revising this section to ensure a smoother flow of the narrative, without the need to constantly refer back and forth.

> Removed the sentence on L144 and generalize this method to the Secant method, as suggested from Reviewer #1.

• L148: You might want to consider giving more intuitive names to Process 1, Process 2, and Process 3, in the text. For example, you could label them as follows: "Step 1: Calculate Correlation Length", "Step 2: Assign Melt Days / Dry Days", and "Step 3: ??", as you do in Figure 3. By the way, it appears that "Step 3" (or Process 3) is not represented in Figure 3.

> Split "processes" into "Step #1: Single $L_{MW}$", "Step #2: Monthly $L_{MW}$", and "Step #3: Daily $L_{MW}$". Process #3 (Now Step #3) is shown in Table 3b*.

This minimization results in modeled microwave grain sizes during potential dry days that, when input into SMRT along with temperature and density from CFM, each produce a brightness temperature that is within 0.1 K of its respective AMSR-2 observation. To invert this radiative transfer model, we run an algorithm with three separate steps: Step #1, Step #2, and Step #3, which are illustrated in Figure 3. Note that we would have found equivalent results by simply following the full algorithm depicted in the flow chart without breaking steps into Steps #1 through #3. We only separate out these processes so that we could initially find general estimates of microwave grain size on fewer days instead of running it on every single potential dry day, thus decreasing the code's runtime.

• L190: What type of 'other information' is meant here?

    Changed to "In the absence of an accurate microwave grain size dataset or model output"

• L191: To maintain consistency, I recommend using 'melt' and 'non-melt' days, as you have done in this line, rather than 'melt' and 'dry' days, which are used elsewhere in section 3.1 and Figure 3.

    Switched to 'dry' days to 'non-melt' days everywhere.

• L198: You could add 'hereafter referred to as 4σ or dynamic threshold).

    Switched to add "or dynamic threshold".

IV. Results

• L215-L222: These lines introduce a significant amount of supplementary information, which may divert attention from the main narrative. Could this be included in the method section rather than the results?

    These lines rely on Figure 4. Since its necessary to keep Figure 4 in the results for validation of correlation length against

• L224: Wouldn't it be more appropriate to introduce SSA in the data section? And also the next line (L225), seems to be more fitting for the method section that for the results.

    Added to new section in Methods called "Section 2.4 Snow microstructure data and concepts" for information on SSA, MOA, microwave grain size, correlation length, etc.

• L245: Consider referring to AWS melt, instead of SEB melt, in the section title.

    Changed to AWS melt.

• L258: What is meant by 'increases across each of the four austral summers shown'? I was expecting the average correlation length per melt season to progressively rise compared to the preceding season, but this trend isn't clearly evident in Figure 5b.

    Thank you for noting this; that is indeed unclear. We changed it on line 258 to reflect that it rises from austral fall throughout austral summer and decreases again during the austral winter for each of the austral summers known.

• L265: Also Torinesi's method yields an accuracy of 91.7%, correct? While you refer to it as the 'second best technique', from my perspective, it actually attains the lowest accuracy (tied with Zwally & Fiegles' method). Could you clarify if I'm overlooking something in this regard?

    Thank you for correcting this. These values are all fairly close together; we are following your recommendation by adding more AWS sites which will provide clarity that these methods produce similar results to one another at the 8 AWS stations.

• L290: As I highlighted in the 'Major comments' section, I believe the manuscript would greatly benefit from including a comparison with additional weather stations. Comparing the Hybrid Method with other statically-based techniques may not provide substantially new insights. Given that statistically-based methods are acknowledged to have limitations in accurately detecting melt (which prompted the development of this new method, as you state in the introduction section), what meaningful insights do we gain from this comparison?

> That is an excellent point, and we have added 6 more sites to gain more insight into the hybrid method's performance.

• L295: I would consistently use capitals for your new method, using 'Hybrid Method' rather than 'hybrid method'. Currently, both are employed (sometimes also in one sentence, for example in L599).

> Switched to Hybrid Method everywhere.

• L301: Could you explain why this is not the case for AWS15?

> Noted that there is a discontinuity in the AMSR-2 data at AWS15 in June 2018. It seems that the Picard method is picking random days during the austral winter and labeling them as melt in that region of the Larsen C, which could be why this isn't the case for AWS15. We plan to present this information in much greater detail in our revised manuscript.

• L309: I would suggest to introduce MOA in the method section.

> MOA has been added to the new section on snow microstructure in Data & Models.

• L303-L322: This appears somewhat abrupt and disrupts the flow of the narrative. Perhaps consider creating a new subsection in the results dedicated to the evaluation of the generated correlation length product?

> We are splitting these up into different subsections.

• L314: I'd be interested in seeing a spatial map that compares MOA's optical grain size with your correlation length product. Additionally, a scatter plot illustrating the correlation between both products would be informative.

> We will be including both of these for the Larsen C ice shelf.

Discussion

• General comment: I recommend using distinct titles for the topics discussed in the subsection to enhance readability.

> We have added subsections accordingly.

• L325: Consider incorporating more specific statistical terminology, instead of mentioning that the methods are 'tied'? This might enhance the clarity of the comparison.

> This is indeed confusing; our goal was to emphasize that our method is hybridized to Picard et al., and not to compare the two technique's accuracy. Therefore we switched "tied" to "hybridized".

• L338: Which two sites do you mean?

> Specified "AWS 17 and 18".

• L339: Can you quantify this statement?

> Specifies that they are accurate within x% (a value we will change upon finishing our assessment of the other 6 sites).

• L363: What do you mean by 'resulting overestimate of emissivity'?

> Changed to "…The resulting overestimate of emissivity due to an underestimate of icy layers within CFM…"

• L379: What do you mean by 'that site'? Are you referring to a specific location or is this a general statement?

> Changed to "between a site's percentage of melt days and microwave grain size".

• L379: Did you test that there was a significant correlation? This might be good to add to the methods section.

> In response to Reviewer #1's suggestion, we clarified that we used Pearson correlation coefficient. The significance is denoted by the $p$ value associated which is listed on L320.

• L399: In my (somewhat limited – sorry about that!) understanding of correlation length, I thought that surface melt could also result in a decrease in correlation length. As the ice undergoes a phase change from solid to liquid, this transition can introduce irregularities in the ice structure, such as the formation of pores, cracks, or fractures filled with meltwater. I thought this process could reduce the correlation length. I assume this is incorrect, right?

> I think a major source of the confusion of the term "correlation length" is that it is not clear from the term its that it's a way to quantify the size of the snow grains within the snowpack. Correlation length can refer to a lot of different physical concepts. Now that we refer to this as "microwave grain size" and include more detail about how it works within the methods section, it should be much clearer in the revised manuscript. When liquid water content is high, grains grow quite quickly, causes refreezing in areas between grains, causes melt crusts. This results in large grain sizes associated with melt.
>
> We just added this citation to help increase readability of this section-
> https://doi.org/10.5194/tc-17-2323-2023

Conclusion - Looks good!

> Thank you.

Author contributions - Shouldn't it be MS instead of CS?

> Adjusted to CMS.

Data availability - Really nice that the data are shared on Zenodo.

> We're glad it's appreciated!

- Could you provide information about the sources from which you obtained all the data products utilized in this study?

> Included information about where data was sourced from.

Figures - Figure 1: Consider adding a title to the legend, indicating something like 'AWS located on:' for clarity. Currently, it might be a bit confusing at first glance.

> Added a title to the legend.

- Figure 1: In the caption, I think 'AIS' should be replaced by 'AWS'. Or, even better, avoid abbreviations in the figure captions, and replace by 'Antarctic Weather Stations'.

> Replaced with "Automatic Weather Stations"

- Figure 1: The shades of blue and cyan appear quite similar to me. Please consider replacing one of the colors?

Replaced the colors.

- Figure 2: Nice figure! One very small comment, maybe you could add one line with a general description at the beginning of the caption?

Thank you. We added a one-line description.

- Figure 3: Nice figure! - Figure 3: Process 3 is mentioned in the text (L148), why is this step not presented in the figure?

Process 3 now clarified to be substep #3 is shown in Figure 3b*

- Figure 3: What are TB1 and TB2? Do they refer to horizontal and vertical polarizations, or the 'low' and 'high' brightness temperatures coming from the 'low' and 'high' correlation lengths?

Yes, in (b) and (c) they are shown as input using $p_{initial}$ + bound and $p_{initial}$ – bound.
Clarified this within Figure 3 caption.

- Figure 4: Again, I struggle to see the difference between blue and cyan.

Changed the colors.

- Figure 4: Could you clarify what is referred to as "NIR SSA" in the caption? Is this data product discussed in the text? Additionally, consider using more intuitive labels, such as indicating the 4 and 10 meter penetration depths, instead of " REFL. SP1" and "REFL. SP2".

Changed to "Snow Pit #1" and "Snow Pit #2".

- Figure 4: In subpanels (c) and (d), I cannot see the AMSR-2 line. Consider increasing the linewidth, so you see it is overlapping with the other lines.

Changed to crosses based on a similar suggestion from Reviewer #1.

- Figure 4: I would suggest not abbreviating 'BC' in the caption, as there are already numerous abbreviations in use, which can hinder readability.

Changed to "Bias correct."

- Figure 4: In panel (c) and (d), the legend states that $-4\sigma_{p_{exp}}$ is shown. Shouldn't this be $+4\sigma_{p_{exp}}$? And in panel (b), both + and – are shown. I got a bit confused here, when to use – (like in panels c and d), and when to use – and + $4\sigma_{p_{exp}}$ (like in panel b)?

By underestimating our microwave grain sizes, we overestimate brightness temperature. Subtracting a value from microwave grain size and putting it back into SMRT results in an overestimate of brightness temperature, which is what we're looking for. We added some clarification about this on L202.

- Figure 4: Please consider merging panels (c) and (d). If they are plotted separately, the distinction isn't clear to me. Alternatively, displaying only panel (d) and (e) could effectively convey the message.

We are merging the panels to show that these values are almost equivalent.

- Figure 4: In panel (a), 18H is depicted, while in panels (c) and (d), 18V is utilized. It is not entirely clear to me why 18H is employed for melt detection, and 18V for establishing the threshold.

We now specify in the methods section that 18H is always used for the Picard et al +/-7 day potential melt days because traditional statistically-based methods do not work for melt detection.

- Figure 4: In panel (b), I find it challenging to discern the median filter. Additionally, could you clarify the purpose of using the median filter?

> Removed the median filter. It was useful for a previous iteration of the hybrid method but not the one we currently present.

- Figure 5 and Figure 6: I'm having difficulty comparing the melt days obtained from the Hybrid Method (panel d) with those derived from the statistically based method (panel a). Could you consider plotting them in a single figure for easier comparison?

> We're replacing this figure with all 8 AWS stations we now compare.

- Figure 7: This figure is not mentioned in the text, please include it in the results section.

> This figure is referenced on L275. We are including a reference to Figure 7b and 7d, as they were not previously included.

- Figure 9: I recommend placing this as the initial figure, possibly in the form of a scatter plot where you plot % melt days against correlation length. This visualization would effectively demonstrate that correlation length is a reliable indicator for surface melt, thereby bolstering your argument for using this method.

> We're adding a new figure where we will incorporate this Figure as a scatter plot in a new figure earlier in the Results section.

Used references - Johnson, A., Fahnestock, M., & Hock, R. (2020). Evaluation of passive microwave melt detection methods on Antarctic Peninsula ice shelves using time series of Sentinel-1 SAR. Remote Sensing of Environment, 250, 112044.

**Review #2**

Review of "A physics-based Antarctic melt detection technique: Combining AMSR-2, radiative transfer modeling, and firn modeling"

The study proposes and evaluates the performance of a new technique to detect meltwater in Antarctica from passive microwave satellite observations. The method is a significant conceptual advance towards a more physics-based estimation of this quantity. Meltwater is of major interest to understand the dynamics of ice shelves (and of the ice sheets). A large corpus of studies use meltwater detected from space and especially from passive microwave observations available from 1979 to day. The potential interest of this study for the snow remote sensing community is very high.

The fact that the method does not outperform the statistical methods is not a surprise, the effect of liquid water on microwave is so strong that a binary detection – melt / no melted – is relatively un-challenging. However, as stated in their conclusion, the potential of a physics-based method is manifold, behind the basic objective of the binary detection. This study is therefore an important step towards a more comprehensive exploitation of microwave data to investigate the liquid water on the ice sheets.

The paper is of high quality, clear, and straight to the point. The method is described with details and the results are abundant. The discussion is excellent. Many minor issues and

suggestions are listed below. Nevertheless, remains a major possible issue. The simulations with SMRT seem to have been run with densities > 450 kg/m3 in some instance, which is inaccurate. Even though the consequences for the results are expected to be minor, because the method is self compensating for inputs and model errors as demonstrated in the study, it is recommended to assess and if possible rerun all the simulations. To avoid this issue to happen again in the future, the reviewer – who is a developed of the SMRT model – has added warnings in the code.

After this issue is addressed, I recommend the publication of the study in The Cryosphere.

> We appreciate your review and consideration of our manuscript. Our sites do have densities above 450 kg/m3, so this is an excellent point you are making. To address this problem, we have since re-run all simulations (including new simulations across the Larsen C Ice Shelf based off comments from Reviewer #1) using the dense snow correction set to auto, meaning we are using IBA run as "air in ice" at densities >450 kg/m$^3$. This reduces our correlation length/microwave grain size values by approximately 20% overall, but in large part does not affect melt detection.

Detailed remarks:

"correlation length". While this term is commonly used and understood by the snow microwave modeling community, it refers to a general mathematical concept that occurs in many domains, and can result in a confusion here for unaware readers, for instance correlation length is also common for rough surface characterization. Furthermore, without specifying the underlying auto-correlation function this term is loosely defined. This is why some authors have used "exponential autocorrelation function" instead. This issue is one of the motivation for the introduction of the "microwave grain size" concept by the reviewer and colleagues in one of the studies cited by the authors. Mathematically, the definition is independent of the form auto-correlation function, and semantically the term is more specific to the length scale relevant to snow volume scattering. These are some of the advantages of this new concept. Nevertheless, this comment is clearly subjective and opinionated. The authors are free to use correlation length.

> We understand with your assessment of the confusion caused by use of the term "correlation length". We decided to switch our terminology from correlation length to microwave grain size for broader scope of our product, and we are adding an extra paragraph of explanation in the text to L89 in the Data & Methods section.

Furthermore, snow microstructure model  we chose  to implement is the exponential snow microstructure model while using IBA within SMRT (SMRT-IBA) because the exponential model considers grain size a single parameter (Picard et al., 2018). Given that we are using an exponential snow microstructure model, this may be referred to as the exponential correlation length ($L_C$). In this paper, we refer to this value as "microwave grain size" ($L_{MW}$), a unifying concept that allows for comparison to other to snow microstructure parameters using other models and techniques (Picard et al., 2022b). Microwave grain size is a length scale for snow microstructure (Picard et al., 2022b). Microwave grain size ($L_{MW}$) is proportional to the Porod length ($L_P$), which relates snow density ($\rho$) and optical grain diameter ($d_{opt}$) in Mätzler (2002; Picard et al., 2022b):

$$L_{MW} = L_C = \alpha L_P = \alpha \, (2/3) \, (1 - \rho/\rho_{ice}) \, d_{opt} \qquad\qquad (1)$$

Equation (1) shows that (a) microwave grain size is proportional to grain diameter, (b) larger optical grain diameter are associated with higher microwave grain sizes, and (c) higher densities are associated with lower microwave grain sizes. Therefore, microwave grain size is a complicated snow microstructural input into SMRT-IBA that relates to both grain diameter and density.

L30 suggestion: "in brightness temperature " → "in brightness temperature timeseries"

Changed to "in brightness temperature time series".

L35. There are works by M. Tedesco using MEMLS (doi: 10.1016/j.rse.2009.01.009). See also Thomas Mote works inverting the scattering signal, which is a simple variant of the approach followed in this paper.

Included reference to Tedesco (2009) on L37 in the Introduction:

considered these physical properties of the snow. In part, this omission is because the properties of snow over the AIS are not well known. An exception to this omission is Tedesco (2009) who developed a physics based, annual threshold for melt detection using the Microwave Emission Model of Layered Snow (MEMLS). However, recent improvements in atmospheric reanalysis, firn modelling, and snow radiative transfer modelling allow us to create a hybridized physics-based technique for detecting melt on a day-to-day rather than annual basis.

L65. There are other recent evaluations with different results, see doi:10.1029/2021GL096599 and doi:10.5194/tc-16-5061-2022. This is particularly important to develop this part in order to understand the representativeness of the effective correlation length obtained by the inverse method.

We note that the penetration depth can be variable and include information from the studies you to which you refer.

microwave radiometer. Colliander et al. (2022) have found the penetration depth to be approximately 40 cm. The maximum depth of detection for this frequency was shown to be around 1-2 m in Picard et al. (2022). Penetration depth can be variable. We use the 12.5 km x 12.5 km daily gridded product of AMSR-2.

**L75: "too fine", can you give some numbers ?**

> When the model is running, it is using layers with thicknesses on the order of millimeters. Typically, the output is interpolated to larger layer thicknesses in post-processing. We edited the text on L75 to reflect this.

**L77-80: More than the computational cost, the accuracy of RT calculations requires that the layers are typically thicker than the wavelength. Here 1cm is just ok, smaller is not advised. This constraint, often overlook when using RT models, may be added here.**

> Specified this constraint on L76-L77:

>> resolution as MERRA-2: 0.5° latitude by 0.625° longitude (Gelaro et al., 2017). The vertical resolution of the daily output profiles from the CFM varies spatiotemporally, and, generally, is too fine for input into the radiative transfer model. Radiative transfer models become inaccurate when layers are too small comparative to the wavelength of the instrument. Therefore, we

**L86. I'd suggest to move the citation for SMRT out of the parenthesis, that is specific to IBA. Maybe also add original refs for IBA and DMRT.**

> Rearranged citations.

**L88. " snow microstructure as a single parameter". Two points:1**

**1) Depending on the usage "snow microstructure" refers to as the geometry of ice/air matrix in general or more specifically to as "grain size" in a more modern / generalized way. To my knowledge there is no consensus and the authors should clarify here, somehow, that they refer to the latter one.**

**In the former case (my preference), snow microstructure is described not only by a length scale but also by many other properties (density/fractional volume, convexity, polydispersity, ...).**

**2) strictly speaking, the choice made here is about the snow microstructure representation (exponential versus sticky hard sphere) and not the electromagnetic model (IBA vs DMRT). Traditionally these two formulations have used different microstructure representations, so the amalgam, but now IBA (in SMRT) can easily be use any microstructure type (not that in the original IBA the exponential auto-correlation function was not presented, only in a latter paper describing MEMLS). I'd suggest to indicate that the "exponential microstructure representation" is used here, which is parametrized by two parameters (density and correlation length/microwave grain size), and for this I'd recommend to cite the original Mätzler 1999 MEMLS paper.**

1) We removed the comment "snow microstructure as a single parameter" and changed it to "grain size as a single parameter".

2) We modified the manuscript to explain that this is why we chose the exponential snow microstructure model, not why we chose IBA within SMRT. Added to L88:

Furthermore, the snow microstructure representation we chose to implement within SMRT-IBA is the exponential snow microstructure model as it considers grain size a single parameter (Picard et al., 2018). Given that we are using an exponential

L99. What is this amount ? Kelvins ? How many ?

Updated this line to specific that this value depends on how we calculate dry snow brightness temperatures and is variable between 5-10 K.

L110. Figure 2c uses densities > 400 kg/m3 in a range where modeling snow as "ice in air" is becoming increasingly wrong (this equally applies to IBA and DMRT and to any snow RT models). The consequence is that Tb are increasing while they should decreases. See doi: 10.5194/tc-16-3861-2022 for details and especially Fig 2. To solve the issue easily, I'd recommend to invert the medium (that is use air in ice) above fractional volume 50% (density > 917/2 kg/m3). A more advanced alternative is to use SCE instead of IBA (see the mentioned paper) but the stability of this more recent and more numerically-challenging method may be an issue. Unfortunately, it means that re-running all the simulations is required if densities >450 kg/m3 are present in the profiles in the upper 1-2 m of the snowpack. I suggest to first assess the proportion of such cases, and second to assess the impact in the worst cases. Then decide. It is likely to impact the estimate of correlation length, but the melt detection won't be affected because the inverse method tends to incorporate all the artifacts into the correlation length, including this modeling artifacts.

Note: SMRT code has been updated on 29 sept 2023:

- a warning has been added to avoid this frequent kind of issue in the future.

- a new argument is available in IBA "dense_snow_correction" to make it easy to apply an automatic (yet imperfect) correction. The user is responsible to activate this option, until positive feedback is received from the community, to make it as a default.

Thank you for making us aware of this issue and adding the warning to SMRT. We switched over to the dense_snow_correction mode set to auto and re-ran all of our sites because many of our density values exceed 450 kg/m3 in the upper 1-2 m of the snowpack. Our new additions to this manuscript – the Larsen C ice shelf from mid-2013 to mid-2014 was also run using dense_snow_correction. We attempted to use scaled symSCE from one of your 2022 papers (doi: 10.5194/tc-16-3861-2022), but we were running into permittivity errors for certain sites so we went with the

imperfect "dense_snow_correction" within IBA. Figure 3c has been updated and now has its expected discontinuity in density at around 450 kg/m3. Below was added to L89 in the Data & Models section. Discussion of the impact of these radiative transfer model uncertainties is included as well.

Within SMRT-IBA, snow is represented as a two-phase medium (assuming no liquid water), with an ice matrix within an air medium being the standard configuration. However, this approximation becomes less accurate at densities greater than 450 kg m$^{-3}$. Therefore, we keep using IBA but switch to a configuration with air bubbles within an ice medium when ice volume fraction exceeds 0.5. We do this by turning on the dense snow correction within SMRT-IBA. This allows us to model the emission and scattering of snow at these higher densities, following an example in Picard et al. (2022). This is not a perfect correction; we considered an alternative model from Picard et al. (2022) that use the strong contrast expansion theory, but the model was not numerically stable for all locations.

L133 "shorter" → lower or smaller. Shorter is used with wavelength... which appears to be longer at 18.7 GHz compare to 89 and 150 GHz used in the cited paper.

Changed to "lower frequency".

L146. I think what you describe is the "Secant method", (e.g. https://en.wikipedia.org/wiki/Secant_method). The Brent method is supposed to be better (Brent, R. P., Algorithms for Minimization Without Derivatives. Englewood Cliffs, NJ: Prentice-Hall, 1973. Ch. 3-4). Comment based on scipy.optimize.brentq documentation.

We now specify on L146 that this is indeed the Secant method. We've already re-run all of our sites and the Larsen C with the Secant method, but we now specify in the discussion section that the Brent method may be faster:

of computation time. Therefore, expanding this technique to be run ice sheet-wide is a computational challenge. A small change to improve speed would be implanting the Brent method instead of the Secant method during our inversion to calculate microwave grain size (Brent, 1973). For these reasons, statistically based techniques are still an important tool for melt

L175. 0.01 K is a very small error compared to satellite accuracy. This could be relaxed to 0.1 K to reduce inversion computational cost.

Thank you for your suggestion. We relaxed this value to 0.1 K, improving computational efficiency.

L180-185. Maybe this paragraph should be earlier, it is indeed difficult to understand why L147 paragraph is needed after L139 paragraph which seems to be a complete and sufficient description of the process. I'd suggest at least to start L139 paragraph with "To further decrease the computation cost ...". If I understand well the purpose of these extra steps.

Paragraph from L180-185 was moved up to L138 and combined with paragraph on L139. Paragraph on L147 was abbreviated.

In Substep #1: Single $L_{MW}$, for a potential dry day, we run SMRT once with an overestimate or a "high" value for microwave grain size and a second time with an underestimate or a "low" value for microwave grain size. We assume temperature and density profiles based on CFM outputs. This results in both an overestimate and an underestimate for brightness temperature, as brightness temperature decreases as a function of microwave grain size at this frequency. Using these two endmembers as bounds, we linearly interpolate to the observed brightness temperature to find a microwave grain size associated with this value. We repeat this process until we converge on a microwave grain size for that day that, when input into SMRT, produces a modeled brightness temperature that is approximately equal to the observed brightness temperature. This process is known as the Secant method (Wolfe, 1959).

This minimization results in modeled microwave grain sizes during potential dry days that, when input into SMRT along with temperature and density from CFM, each produce a brightness temperature that is within 0.1 K of its respective AMSR-2 observation. To invert this radiative transfer model, we run an algorithm with three separate substeps: Substep #1, Substep #2, and Substep #3, which are illustrated in Figure 3. Note that we would have found equivalent results by simply following the full algorithm depicted in the flow chart without breaking steps into Substeps #1 through #3. We only separate out these processes so that we could initially find general estimates of microwave grain size on fewer days instead of running it on every single potential dry day, thus decreasing the code's runtime.

First, we estimate a general microwave grain size for that location for Substep #1: Single $L_{MW}$. Then, estimate microwave grain sizes monthly for Substep #2: Monthly $L_{MW}$. Since Substeps #1 and #2 are purely estimates, we only consider the steps in Figure 3a to 3d one time through for each of these two processes. Finally, in Substep #3: Daily $L_{MW}$, we converge on a microwave grain size that, when input into SMRT with CFM input temperature and density, best matches the local observed brightness temperature.

**L208. I'd remove "more"**

Removed "more".

**L219. Please add the value for the bias.**

Please see next comment where we add the linear regression equation.

**L219. "remove it by linear regression". Why not by subtraction ? Is this bais has a seasonal variability ? Give details about this linear regression, in short.**

On L220 we add details of the linear regression:

temperature to in situ temperature data from Dome C to check for a seasonal bias. Indeed, there is a seasonal bias in the MERRA-2 skin temperature for this site, and we remove it using a linear regression. The relationship between the corrected temperature ($T_{corrected}$) and air temperature in CFM is: $T_{corrected} = T_{CFM} * 1.066 - 9.246$. We re-run the CFM with the bias-

Fig 4. I'd expect to see AMSR-2 data in Fig 4c and Fig 4d. Do they overlap ? If yes, maybe use small crosses for the AMSR-2 data.

> They do overlap almost exactly; therefore, we switched to small crosses for the AMSR-2 data so that both lines are visible to the reader.

I think the data you are using are coming from Picard et al. 2014, not from Brucker et al. (2011) which was limited to 2m in depth. I don't understand "Refl." in the legend in Fig b.

> Thank you for catching this mistake, we have corrected it to cite Picard et al., 2014. Removed "Refl." In legend, changed to "Snow Pit #1" and "Snow Pit #2".

L228. penetration depth of AMSR-2 → add "at 18 GHz"

> Changed to "19 GHz" (based additionally off comment by Reviewer #2 to round 18.7 GHz to 19 GHz).

L234 Figure 3e → Figure 4e. In addition I'd suggest to remove this last panel, the value of 0.02K in the text is sufficient for the reader to understand that it is extremely small and negligible.

> We removed the last panel (Figure 4e). We also updated the difference to be 0.1 K rather than 0.01 K following your suggestion to relax the +/- 0.01 K requirement for the inversion.

Fig 7. To avoid overlapping symbols, I'd suggest to use thin vertical bars as symbols in the panels b and d.

> Corrected, we now use thin vertical bars for Figure 7b and 7d.

L308. The Mosaic was referred before in the paper as an abbrevation and without reference. This should be corrected.

> The original reference to MOA in the Data & Models section was updated to include the name and the reference in L308 was switched to MODIS Mosaic of Antarctica.

L310 and L320. I'm not sure what kind of mathematical correlation is used here.

> Updated to reflect Pearson correlation coefficient.

L340. It is nice to recall that the threshold proposed in Picard et al. 2022 was not intended to be optimal. It does surprisingly well.

Yes, that was an interesting finding for those locations. Although with the addition of 6 more AWS sites in response to Reviewer #1, it looks like Picard et al. does not perform much better than Torinesi et al. (2003)

L368. This final part of the discussion should be developed a bit. Retrieving correlation is a useful side product of this work, but the representativeness of the value is a big issue. In particular, because there is a circular indetermination between the correlation length and the penetration depth. Usually large correlation lengths imply low penetration depths (for a given density) which means that the values obtained here are representative of depths that depends on the values.

Clarified on L402 in the Discussion:

snow microstructure influences radar penetration depth. However, there are limitations to utility of the microwave grain size product of the hybrid method as it currently stands. Our calculations of microwave grain size refer to a bulk layer with a depth that depends on penetration depth of AMSR-2 at that frequency. Indeed, the penetration depth at a given frequency and snow density also depends on the microwave grain size. This circuitous relationship means that representativeness of our microwave grain sizes to the physical snowpack using the hybrid method is currently a problem. However, future developments such as using two different frequencies to establish an approximate surface microwave grain size and slope with depth, similar to techniques used in Picard et al. (2012), could help alleviate this issue.

L370. It is worth mentioning that the SEB estimates are not pure observations, a model is involved. Despite the relatively weak assumptions required to compute melt for the raw meteorological observations, some assumptions are required.

Specified the uncertainty of using SEB estimates on L377 in the Discussion.

associated with localized melt events that are nearly averaged out by the 12.5 x 12.5 km footprint of AMSR-2. Finally, these SEB-derived melt data contain some uncertainty associated that is both measurement errors as well as model accuracy (Jakobs et al., 2019).

L390 in MERRA-2 → consider to add "and in SMRT".

Added "and in SMRT".

Discussion: I suggest to add a few items regarding scalability of the technique at the continental scale. It could be mainly about computation time as I don't see any other difficulties.

Added discussion of the issue of scalability for hybrid method on L403 in the Discussion:

days versus dry days. An important aspect of the hybrid method that is worth noting is that is highly computationally expensive. Running SMRT multiple times to complete our inversion for each day and each AMSR-2 grid cell requires a significant amount of computation time. Therefore, expanding this technique to be run ice sheet-wide is a computational challenge. A small change to improve speed would be implanting the Brent method instead of the Secant method during our inversion to calculate microwave grain size (Brent, 1973). For these reasons, statistically based techniques are still an important tool for melt detection.

---

## Referee Report (RR1)

**Review of Dattler et al. 'A physics-based Antarctic melt detection technique: Combining AMSR-2, radiative transfer modeling, and firn modeling'**
By Sophie de Roda Husman ([S.deRodaHusman@tudelft.nl](mailto:S.deRodaHusman@tudelft.nl))

The authors have made significant efforts to enhance the manuscript, resulting in a noticeable improvement in readability. Congratulations on this achievement! At this stage, I have only a few comments remaining, primarily related to some textual changes and the figures.

Here's a brief list of suggestions for the authors to consider:

**Abstract**
- **L11:** Replace "*Snow Radiative Transfer model*" by "*Snow Microwave Radiative Transfer model (SMRT)*"
- **L11:** Replace "*Community Firn Model*" by "*Community Firn Model (CFM)*"
- **L12:** I am not sure what this sentence means (specifically, I don't understand the word "hybridize" in this context), but this could also be because I am a non-native English speaker.
  *"In the process, we also **hybridize** our method to statistical techniques …"*
- **L14:** In the rest of the manuscript, you use a hyphen in "statically based", add here as well.
- **L19:** Add comma: *"… Antarctic snow, and (b) …"*

**Introduction**
- **L36-37:** Consider adding a reference (e.g.: Hofer & Mätzler (1980); Mote & Anderson, 1995)
- **L57:** For people not familiar with microwave sensors, it might be good to introduce AMSR-2. For example by adding some information to this sentence: "…*microwave radiometer Advanced Microwave Scanning Radiometer 2 (AMSR-2)…*"
- **L58:** I think it is "*Automatic Weather Station (AWS)*" instead of "Antarctic Weather Station (AWS)".
- **L59:** Suggestion to replace "*Antarctic sites*" by "*AWS*"
- **L60:** "*Other techniques*" sounds a bit vague. What do you mean by this?

**Data & Models**
- **L63:** Could you replace this by "*thirteen AWS*". Or are the three "dry sites" not automatic weather stations?
- **L64:** analyses (instead of analysis)
- **L83-85:** Please double-check the abbreviations. Once you have introduced abbreviations (e.g., CFM, AIS), for consistency, continue using these instead of the full names.

**Methodology**

- **L58:** Physically-based

**Results**

- **L340:** What do you mean with this sentence: "*This likeness is not the case for the pattern of melt end dates*"?

**Discussion**

- **L402:** "*19H*" instead of "*18H*"
- **L419-429:** Very interesting paragraph. I had some similar findings in a paper where I compared different statistically-based methods for melt detection (de Roda Husman, et al., 2022).
- Really nice and well structures discussion!

**Conclusion**

- **L533:** I thought *thirteen* sites?

**Figures**

- **Figure 1:** Replace "*AIS*" by "*AWS*" in figure caption
- **Figure 2:** Consider rewriting the figure caption, with a first sentence that describes main idea figure, and summarize the rest of the text. Same holds for **figure 4, figure 7, figure 8, figure 10, figure 11, figure 12.**
- **Figure 3:** Shouldn't "*Calculate correlation length*" be replaced by "*Calculate microwave grain size*"?
- **Figure 4:** I don't see the red area in (c), is this missing?
- **Figure 7:** I would use a sequential color palette for the melt duration instead of a diverging one. The white color (around 65 days) seems to have no melt now (if you quickly look at the figure), which is a bit confusing. Same holds for **figure 12**.
- **Figure 8:** In (b) and (d), you cannot see the Picard et al. melt days if the Hybrid Method shows melt. I would suggest to use a symbol for days where both method show melt, or make Picard et al.'s symbol a bit larger, because now it seems that on days where the Hybrid Method shows melt, Picard et al's method does not.
- **Figure 9:** Consider adding the Pearson Correlation Coefficient to (b). Same holds for **figures 10-12.**
- **Figure 10:** The variation in color bar limits is somewhat confusing. Would you consider standardizing the limits to enhance clarity?

**Used references**

- T. L. Mote and M. R. Anderson (1995), "Variations in snowpack melt on the Greenland ice sheet based on passive-microwave measurements," Journal of Glaciology, vol. 41, no. 137, pp. 51–60.

- Hofer, R., & Mätzler, C. (1980). Investigations on snow parameters by radiometry in the 3-to 60-mm wavelength region. Journal of Geophysical Research: Oceans, 85(C1), 453-460.
- de Roda Husman, S., Hu, Z., Wouters, B., Munneke, P. K., Veldhuijsen, S., & Lhermitte, S. (2022). Remote Sensing of Surface Melt on Antarctica: Opportunities and Challenges. IEEE Journal of Selected Topics in Applied Earth Observations and Remote Sensing.

---

## Author Response (AR2)

Dear Stef Lhermitte,

We thank you and Reviewer 1 (R1) for the helpful reviews of this paper. We correspondingly added all of the below recommended edits in the new version of the paper.

Thank you,
Marissa Dattler, Brooke Medley, and C. Max Stevens

**Abstract**
- **L11:** Replace "*Snow Radiative Transfer model*" by "*Snow Microwave Radiative Transfer model (SMRT)*"
- **L11:** Replace "*Community Firn Model*" by "*Community Firn Model (CFM)*"
- **L12:** I am not sure what this sentence means (specifically, I don't understand the word "hybridize" in this context), but this could also be because I am a non-native English speaker.
  *"In the process, we also **hybridize** our method to statistical techniques …"*
- **L14:** In the rest of the manuscript, you use a hyphen in "statically based", add here as well.
- **L19:** Add comma: *"… Antarctic snow, and (b) …"*

**Introduction**
- **L36-37:** Consider adding a reference (e.g.: Hofer & Mätzler (1980); Mote & Anderson, 1995)
- **L57:** For people not familiar with microwave sensors, it might be good to introduce AMSR-2. For example by adding some information to this sentence: "…*microwave radiometer Advanced Microwave Scanning Radiometer 2 (AMSR-2)…*"
- **L58:** I think it is "*Automatic Weather Station (AWS)"* instead of "Antarctic Weather Station (AWS)".
- **L59:** Suggestion to replace "*Antarctic sites*" by "*AWS*"
- **L60:** "*Other techniques*" sounds a bit vague. What do you mean by this?

**Data & Models**
- **L63:** Could you replace this by "*thirteen AWS*". Or are the three "dry sites" not automatic weather stations?
- **L64:** analyses (instead of analysis)
- **L83-85:** Please double-check the abbreviations. Once you have introduced abbreviations (e.g., CFM, AIS), for consistency, continue using these instead of the full names.

**Methodology**

- **L58:** Physically-based

**Results**

- **L340:** What do you mean with this sentence: "*This likeness is not the case for the pattern of melt end dates*"?

**Discussion**

- **L402:** "*19H*" instead of "*18H*"
- **L419-429:** Very interesting paragraph. I had some similar findings in a paper where I compared different statistically-based methods for melt detection (de Roda Husman, et al., 2022).
- Really nice and well structures discussion!

**Conclusion**

- **L533:** I thought *thirteen* sites?

**Figures**

- **Figure 1:** Replace "*AIS*" by "*AWS*" in figure caption
- **Figure 2:** Consider rewriting the figure caption, with a first sentence that describes main idea figure, and summarize the rest of the text. Same holds for **figure 4, figure 7, figure 8, figure 10, figure 11, figure 12.**
- **Figure 3:** Shouldn't "*Calculate correlation length*" be replaced by "*Calculate microwave grain size*"?
- **Figure 4:** I don't see the red area in (c), is this missing?
- **Figure 7:** I would use a sequential color palette for the melt duration instead of a diverging one. The white color (around 65 days) seems to have no melt now (if you quickly look at the figure), which is a bit confusing. Same holds for **figure 12**.
- **Figure 8:** In (b) and (d), you cannot see the Picard et al. melt days if the Hybrid Method shows melt. I would suggest to use a symbol for days where both method show melt, or make Picard et al.'s symbol a bit larger, because now it seems that on days where the Hybrid Method shows melt, Picard et al's method does not.
- **Figure 9:** Consider adding the Pearson Correlation Coefficient to (b). Same holds for **figures 10-12.**
- **Figure 10:** The variation in color bar limits is somewhat confusing. Would you consider standardizing the limits to enhance clarity?

**Used references**

- T. L. Mote and M. R. Anderson (1995), "Variations in snowpack melt on the Greenland ice sheet based on passive-microwave measurements," Journal of Glaciology, vol. 41, no. 137, pp. 51–60.

Minor comments by editor:
- L31+51+59+... "statistical techniques" vs "statistically-based techniques" vs "statistically-based thresholding techniques" vs "statistically-based melt detection techniques" vs "statistical thresholding techniques". All these terms are used interchangeably, but it would be beneficial to use one common term throughout the manuscript. As an editor I prefer "statistical thresholding techniques" as they contain the most info + are most concise.
- L63: "We run our melt detection technique for ten AWS and three additional dry sites".
* The difference between AWS and additional sites is not very clear as the description is not very clear. How was the melt determined on the additional sites where no AWS was present? Is the difference between them that they are wet vs dry or are the AWS sites also dry/dry sites also with a AWS? Later on it becomes clear that the dry sites are used separately for validation of the temperature + grain size, but this is not clear when describing the sites
* Why do you mention three dry sites when it seems only Dome C is actually used in the manuscript? The others are shown in the manuscript but never referred to so it remains unclear what their role in the paper is.
- L94 "merge": How was this merging done? Based on which criterium? How were the characteristics of the merged layers determined?
- Figure 5e + Figure 6 + L311-316: It would be beneficial to add the melt fraction (in %) at the right hand side of the figure so the reader can directly see the percentages without having to look them up in Table 1.
- L365 "2018). The Pearson correlation coefficient between these two grain size variables is 0.88 (p < 0.01)." Refer to Fig. 9b + add other common statistics like RMSE, slope etc. Moreover, I think it would be beneficial to add the regression line and 1:1 line to Fig. 9b.
- Figure 7+12: some panels use a diverging color palette whereas the data are not diverging. This may lead to misinterpretation etc. Therefore, check "The misuse of colour in science communication" https://www.nature.com/articles/s41467-020-19160-7 and specifically Fig.5+6 for better choice of color palettes.
- L377: "we can see". Refrain from subjective phrasing
- L379-380 "On Dronning Maud Land there is moderate durations of melt and moderate microwave grain sizes, and on the Antarctic Peninsula there is higher durations of melt and higher microwave grain sizes" -> there are + longer durations + larger grain sizes
- L404 + L442"computational efficiency" I think it is important to clarify for the reader the computational impact. Indicating how long does it take to run the hybrid model on what type of computer setting, would help the reader to better understand the computational impact.
- Supplement: what is the role of the supplement in the paper (?) as it seems nowhere in the paper there is any reference to the supplementary material.